# Flowing between gongs: Mixed-methods insights into shared flow and temporal distortion in music performance

Hannah J. Gibbs ⓘ *, Andrea Schiavio ⓘ

Music, School of Arts and Creative Technologies, University of York, York, United Kingdom

* hannah.gibbs@york.ac.uk

## Abstract

Ideas of temporal distortion prevail in the discourse of flow research, where references are often made to "time flying". Nonetheless, little research has investigated how flow affects time perception in terms of both directionality and surrounding context, particularly within shared flow. Simultaneously, temporal distortion during music listening has been explored, but little is known about how time is experienced by performers. With this in mind, we aimed to investigate whether time distortion is associated with the experience of shared flow state, and the awareness participants have towards influencing factors on such experiences, in the context of music performance. Four groups of Javanese gamelan ensembles (total $N$ = 36, age in years $M$ = 44.83, $SD$ = 14.993, 47.2% female), played in three conditions. We collected qualitative and quantitative data; focus groups and follow-up surveys explored understandings of shared flow and time, while questionnaires included pre-validated scales and an item requiring participants to estimate how much time they thought had passed. Qualitative and mixed methods findings suggest an optimal middle ground of conditions for "time flying" to occur, akin to flow state. Meanwhile, quantitative results indicate a complex relationship between temporal distortion and shared flow, whereby the relationships are opposed under two shared flow factors.

## Introduction

### Introducing the problem

How do you experience time when you are truly absorbed in a task or activity? Say, for example, an entire Sunday passes by while you are crafting or playing a new video game, and suddenly it is dark outside. Or what about a compulsory work meeting, where you feel some contempt towards the chair and one hour feels more like three? These examples capture the implications of enjoyment and attention on time perception, and in turn why time distortion is such a predominant feature of flow state.

Nevertheless, something missing in the recent emergent interest in the state of flow is an investigation as to whether this quality of time distortion is shared amongst a group in flow together, during experiences of shared flow. The present research, therefore, aims to examine

**Data Availability Statement:** All datasets, analysis code and focus group transcripts are included in our OSF repository, https://osf.io/tv7x9/.

**Funding:** At the time of writing, the first author is under receipt of AHRC Open-Competition Doctoral

funding through the White Rose College of Arts and Humanities, grant number AH/R012733/1. The funders had no role in study design, data collection and analysis, decision to publish, or preparation of the manuscript.

**Competing interests:** The authors have declared that no competing interests exist.

temporal distortion during shared flow in a group music-making context. This may be complex; as music exists in time, there is a potentially confounding experience between the experience of time as a result of flow, and the experience of time as a result of music itself. And, while a state of flow might be characterised by a degree of subjective time distortion, this may be difficult for performers to specify if asked to estimate.

By combining quantitative measures with qualitative insights, we aim to ascertain the degree to which the time distortion is associated with the experience of shared flow state, and the awareness musicians have towards influencing factors on time and shared flow.

**A brief history of time perception.** In Western philosophy, early accounts of perceived time are thought to have begun with Aristotle, who defined time as sense perception, undefinable by itself; rather, it is a number in motion, grounded on a perception of the before, and after [1]. Later, philosophers paid greater consideration towards the contextual factors we alluded to before. Locke noted that through intense absorption and focus on a singular object or instance, the resultant perception of time passed is short [1]. Conversely, Thomas Reid described how if such levels of focus were given to the perception of pain, the perception of time is lengthier, and it is the experiences of joy and cheerfulness in which time is shortest [2].

Such notions of perceptual and objective differences were well-defined by the end of the 19th century, in the work of Henri Bergson in 1889. For him, time is divided into two meanings: objective, scientifically measurable time, referred to as *temps*, and subjective, unmeasurable, perceived time, or *dureé* [3]. *Dureé* is not based on *temps* but on a continuous flow of subjective durational experience. Reconciling Aristotle's notion of motion, with Thomas Reid's and Locke's ideas on experiential and contextual influences, Bergson argued that perceived time is guided by emotions, memories, and worldly perceptions. With references to 'lived time', Bergson's ideas are seen as phenomenological precursors. And, like Bergson, phenomenological philosopher Edmund Husserl saw time as both complex and subjective, where temporal sensemaking was dependent on objects of consciousness. His theory of internal time consciousness describes subjective temporal experiences as an interplay between phases, namely between past, future, and present [4].

Fast forward, and we have a clearer picture of contextual factors influencing our perception of time, though one that is not too dissimilar from Reid's, Locke's and Bergson's views outlined above. Such factors usually include, but are not limited to, extensive concentration and enjoyment [5], attention and memory [6], and emotional states [7–9]. When it comes to music, Knowles [10] suggests two cognitive facets to temporal distortion during a period of listening. The first is that of complexity, whereby higher complexity may result in temporal contraction, and vice versa. However, the relationship is not linear, and subsequently excess of complexity may lead to greater overwhelm. The second facet is that of attention; in instances in which attention is distracted from temporal aspects of music, subjective perception of time is contracted, and where our attention is directed towards time, it is expanded. This relates to earlier theories of Locke and Thomas Reid [2], and although the discussion is entirely centred on the experience of the listener, one can see how it is relevant to the experience of music performance.

**Flowing through time within Javanese gamelan performances.** When discussing time perception, experience, and contextual factors such as absorption, attention, pleasure, and complexity in relation to music, the notion of "flow state" might immediately spring to mind as the common link. Initially conceived by Csikszentmihalyi, it typically encompasses nine factors, including balance of challenge and skill, concentration, awareness, and transformation of time [11, 12]. However, in most methods of measuring flow, time perception is usually only measured by arguably loose and ambiguous references to losing track of time [13, 14], and few studies have explicitly assessed the relationship between flow and time perception using a

quantitative measure of time. Those that have incorporated a quantitative measure have focused on one particular facet of flow, typically via moderate attentional demands, challenge-skill balance, or optimal concentration, whereby evidence of subjective temporal contraction has been acknowledged [15–17]. However, little attention has been paid towards other facets of flow, or shared flow, and their relative consequences for time perception.

In the context of music listening or in tapping studies, time perception has had some attention [18–21], but little in the context of naturalistic music performance. One obvious reason for this is that for those who involve themselves in musical performance, time bears little importance. Nevertheless, it may be fruitful to consider whether congruity of the direction and the extent of time distortion in the context of group music performance is associated with self-reported shared flow state, and influenced by the same factors that may influence flow.

The focus of this study is on Javanese gamelan performance, an inherently ensemble-based set of predominantly metallophones and resonant percussion instruments struck with mallets, where traditional pieces encompass interlocking and reciprocating streams of melody and rhythm repeating in cycles [22]. In traditional pieces, known as *karawitan*, basic melodic lines are played by keyed instruments under the bracket of the *balungan*, while the structure is played by gong-type instruments. Other elaborating instruments can be added, including the *suling*, a wooden flute, *rebab*, a two-stringed bowed instrument, and *gender*, a keyed instrument played with two soft mallets, all of which play more complex patterns with some variation [23]. Through a process referred to as *garap*, these complex patterns, or *cengkok*, are constructed in accordance with the decisions of other members of the group and *rasa* [22]. *Rasa* roughly refers to the "feeling" of a piece, akin to Western conceptions of musical affect, though is near impossible to directly translate, and is collectively established and maintained by a group in a similar manner to that of shared flow [24].

This process of *garap* underlying *karawitan* is often what other scholars have referred to as what gives gamelan an "improvisatory character" [25], and while not traditional *per se*, gamelan lends itself well to more free forms of group improvisation as a result of the instruments being harmoniously tuned to one another [26]. More generally, *karatiwan* can be learnt through notation, or memorised with little challenge and are usually led by a drummer. However, unlike the role of Western orchestral conductors, their role is largely to maintain the *rasa*, nurture a sense of cooperation and minimise hierarchy [27, 28], all the while providing structural cues and guiding changes in tempo and density. Such changes are closely linked in gamelan music under the bracket of the term *irama*, which concerns both time and space. In the simplest of examples, as the tempo may decrease from one variation into another, many instruments are required to double the density of their material; the result is that the tempo is roughly the same in actuality, but a variation would take twice as long to perform [24, 29].

Loose attitudes towards time are intrinsic within Indonesian society. *Jam karet*, meaning "rubber time", is used in everyday language to describe how time expands and shrinks, akin to that of rubber, and reflective of a laid-back societal belief [30]. Issues of timing in gamelan itself have had some attention due to the simultaneous tensions on coordination and flexibility [31, 32]. Meanwhile, shared flow itself is deeply engrained within gamelan, through the concept of *ngeli*, interpreted as flowing together and being carried by the music [28]. Extending these ideas beyond *rasa*, *irama*, musical metre, and metric timing, towards ideas of time distortion and flow in gamelan is somewhat inescapable.

## Aims

This paper aims to further disentangle the time-related aspects of shared flow through two studies, by investigating participants' awareness of the influencing factors on time and shared

flow, and testing whether there is a relationship between subjective time distortion and shared flow state. As such, Study 1 concerns an exploration of participants' views of flow state and time in gamelan performance, and the reasons or influencing factors they could attribute to their experience of shared flow or time. For Study 2, we were then interested as to whether time distortion as a standalone measure was associated with antecedents of shared flow and shared flow factors, and contributing factors outlined by participants, in the context of gamelan performance. Finally, through mixed methodology, we wanted to compare qualitative reasoning underlying time perception and shared flow with quantitative findings.

## General method

### Overview

We present two studies based on the same procedure, qualitative and mixed-methods respectively, which was repeated for four pre-existing gamelan groups. This paper is part of a larger project, and therefore certain aspects of the procedure and methods, including participants, are the same as another currently under review [33]. Due to distinct research questions and methodologies, we have maintained the papers as separate endeavours. While we adopted a convergent design during data collection, the order in which we present the studies was selected on the basis of informing our analytical choices. Study 1 consisted of focus group interviews, where the aim was to gain insights into participants' views of the temporal and flow-related experiences after playing gamelan, and how such experiences could be accounted for through contributing factors. These contributing factors then informed the analysis of Study 2, which relied on the questionnaire and individual follow-up survey data, collected during and after the experiment respectively. All data and accompanying code are available at Open Science Framework (https://osf.io/tv7x9/).

### Procedure

Fig 1 provides an overview of the chronology of the procedure, and how the data is presented in this paper. The overall procedure consisted of a repeated measures design, whereby each group was invited to perform traditional gamelan pieces in their usual rehearsal space from notation and memory, and an improvisation.

Participants were asked to remove watches, and any clocks and phones were kept out of sight, which participants were told was to avoid distractions. The performances were interspersed with questionnaires, while physiological measures were recorded throughout; these physiological data are the primary focus of our other paper—based on a different research question—currently under review. Immediately following the final questionnaire, a focus group interview was carried out with each group while their experiences were fresh, and a follow-up personalised survey was administered to all participants to complete individually several days later.

### Participants

Table 1 provides demographic information for all participants. A total of 36 participants across four groups were recruited to participate in the study, between 23rd June 2023 and 2nd August 2023. Ethical approval was given by the University of York Arts and Creative Technologies Ethics Committee prior to recruitment. Participants provided written consent, and were informed that they may withdraw from the research at any time, that their individual participation would be treated anonymously, and that the data would only be used for research purposes.

Procedural order:

**Fig 1. Procedural design.** Solid lines indicate the order data collection and reporting, while dotted lines indicate how each category of data collected are incorporated into the two studies we report in this paper.

As with a previous study conducted by the first author and colleagues [34], the first group consisted of 12 members of Gamelan Sekar Petak (GSP) a gamelan group based at the University of York. This group consisted of several students who had only begun playing at the start of the year, others who had played throughout their degree and were in the final stages, and three long-standing members of the group, who had many years of experience playing and teaching gamelan playing. However, all members studied, or had studied music and had a moderately high level of musical training according to the GOLD-MSI musical training sub-scale [35]. Repertoire typically consists of traditional and more modern gamelan pieces, as well as new compositions created by members of the group.

The second group involved seven members of Southbank gamelan players (SBGP) a group based in London. The group consisted of highly experienced gamelan players, most of whom had spent some time learning techniques in central Java, though had a moderately high level of musical training according to the GOLD-MSI. Since the COVID-19 pandemic and because of other commitments, they have met more irregularly than most groups, but typically perform pieces from memory.

**Table 1. Demographic information for each group, and across the total participant sample.**

| | | Gamelan Sekar Petak | Southbank Gamelan Players | Dublin National Concert Hall Gamelan | Gamelan Spréacha Geala | Total |
|---|---|---|---|---|---|---|
| **Sample** | N | 12 | 7 | 8 | 9 | 36 |
| | Male | 6 | 4 | 6 | 3 | 19 |
| | Female | 6 | 3 | 2 | 6 | 17 |
| | Ages (years) | 20–45<br>29.17 ± 8.16 | 39–67<br>54.71 ± 9.98 | 33–67<br>46.75 ± 11.09 | 42–75<br>56.33 ± 11.06 | |
| **Training** | N reports | 12 | 6 | 8 | 9 | 35 |
| | **Musical training (GOLD-MSI)** | 20–38<br>31.08 ± 4.98 | 26–38<br>33.33 ± 4.37 | 26–42<br>33.88 ± 5.67 | 12–37<br>26.33 ± 7.78 | |
| | **Gamelan experience (years)** | 0.8–25<br>7.42 ± 9.69 | 18–43<br>33.43 ± 9.13 | 4–20<br>8.88 ± 5.08 | 1.5–24<br>7.57 ± 6.46 | |

Musical training was calculated as the sum of the musical training factor of the GOLD-MSI. Years and training scores are indicated as a range with mean ± SD.

The third group involved eight members of Dublin's National Concert Hall Gamelan (NCH). The group is initially a community group, though most of its members have been involved for several years, while others had recently been invited from the level two group. This group also had the highest average level of musical training, though this was more variable in comparison to SBGP.

The fourth and final group involved nine members of Gamelan Spréacha Geala (GSG), a community group based in West Cork. The group in general had the least amount of gamelan playing experience, the lowest average level of musical training and the greatest variability across members, but typically meet on a weekly basis.

## Study 1

### Method

**Focus group interviews.** A semi-structured focus group interview was conducted by the first author for each group after the experiment in the rehearsal space, each lasting around 25 minutes. There were three broader themes underlying the questions asked. Firstly, participants were asked to provide their understanding of flow state and group flow theories, before they were clarified to the group. Secondly, participants were asked to contextualise these theories in their playing during the experiment and discuss factors that may influence their experiences of flow. Lastly, participants were asked to consider the role of time perception in their own experiences of flow. Overall, the purpose of the focus group was to gain subjective insights that could inform the analysis of quantitative data. As gamelan is intrinsically a group activity and the focus of the research is on group flow, the approach of focus groups, rather than individual interviews, seemed more applicable to the context. All focus groups were around 25 minutes in length, conducted on the same day as the experiment, and recorded using a Zoom H4nPro. All recordings were later transcribed verbatim by the first author. Transcripts are included in our OSF repository, while the focus group schedule is provided in S1 File.

### Data analysis

**Focus group analysis.** We implemented a hybrid approach to analysing focus group data, which involved cycles of deductive and inductive techniques. Prior to coding, *a priori* codes and themes were generated in line with the aims of the study, by means of a deductive approach. The broader categories of these were the following:

○ Perspectives on flow and shared flow

○ Subjective time and musical time

○ Contextual influences on flow and time

Using MAXQDA software, an inductive approach was then adopted, where data were reviewed line-by-line and coded in vivo. The in vivo codes were then refined and categorised for each focus group separately, and with deductive categories in mind, while still being open to the possibility of unexpected insights. This allowed us to stay grounded within our aims while being fully immersed in the data. Quotes and excerpts are provided with both group acronyms and individual anonymous identifiers (the letter P and a number following the order of speakers within each focus group). This allowed us to contextualise our interpretations within each group's unique context. Code interrelations were explored visually using MAX Maps [36], inspired by the recent work of Kruse-Weber and co-workers [37].

**Findings.** The iterative cycling of coding focus group transcripts led to five main categories, representing the main areas of enquiry, grouped into two overriding themes:

1. Primary understanding and experiences

- experiential qualities of flow

- group aspects of performance

- temporal experiences and estimations

2. Secondary influences and outcomes

- obstructions, hindrances, and instability

- factors and antecedents.

An extensive list of codes emerged from the data and were grouped into each category, detailed in Fig 2.

In this section, we explore each of the first three categories separately under the theme of "Primary understanding and experiences", while also discussing co-occurrences with any other codes. The final two categories under the theme of "Secondary influences and outcomes", were primarily of interest in terms of where codes co-occurred with other categories and are therefore largely discussed in tandem. However, we also explore the intersections between the two underlying categories of this theme *obstructions, hindrances, and instability*

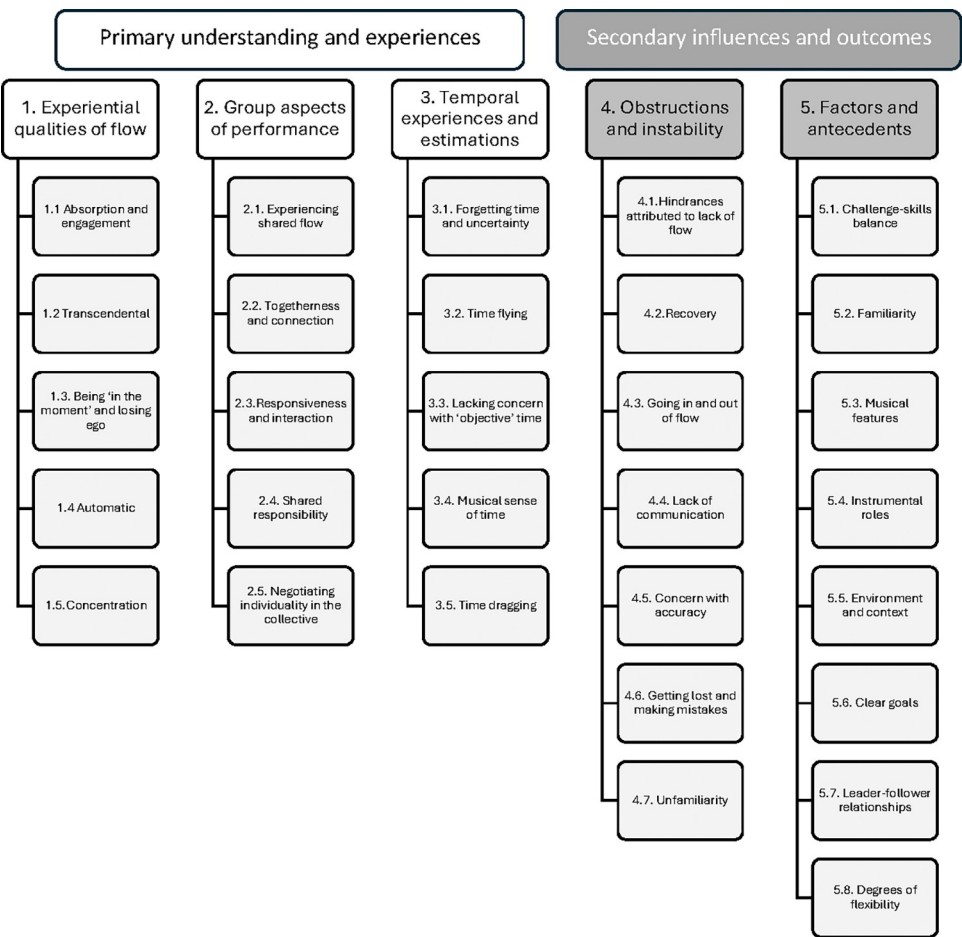

**Fig 2. Coding scheme from focus group interviews.**

*(4)*, and *influencing factors and antecedents (5)* to gain a more complete picture of how the two categories overlap.

**Primary understanding and experiences.** *1. Experiential qualities of flow.* When players were first asked about their understanding of what flow state is, several made immediate reference to Mihaly Csikszentmihalyi, suggesting prior knowledge of flow theory. Greater interest from general audiences in the phenomenon seemed to have stemmed largely from growth in popular culture and social media. Regardless, fascinating insights were presented when participants were invited to describe their understanding of flow state. Fig 3 presents code interrelations between the codes under this category and codes of any other category.

Distinctions were often immediately made between characteristics of flow experience itself and the antecedents for flow to be achieved. The former commonly encompassed absorption and losing track of time, while the latter concerned prerequisites that must be in place in order to experience flow. Consider the following quote:

> "My understanding of it is it's something that you experience doing many different things [. . .] where you become absorbed, and you lose a sense of time. But there has to be a certain level of competence involved". (NCH, P5)

Here, the potential for flow to be achieved is attributed to the antecedents of competence and *challenge-skill balance* (5.1) required, leading to a sense of *absorption and engagement (1.1)* in the moment, which they associated with the emergent property of *forgetting time and uncertainty* (3.1). From this perspective flow is something that you can achieve and experience providing the necessary criteria are fulfilled.

One member of the same group described their experience of flow outside of gamelan in a more out-of-body sense, coded as *transcendental* (1.2): "it was this weird feeling that the music was playing you, that was really strange (NCH, P8)". The distinction here is the potential control one can have over whether flow can be *achieved*, or whether flow *arises*, and in turn the

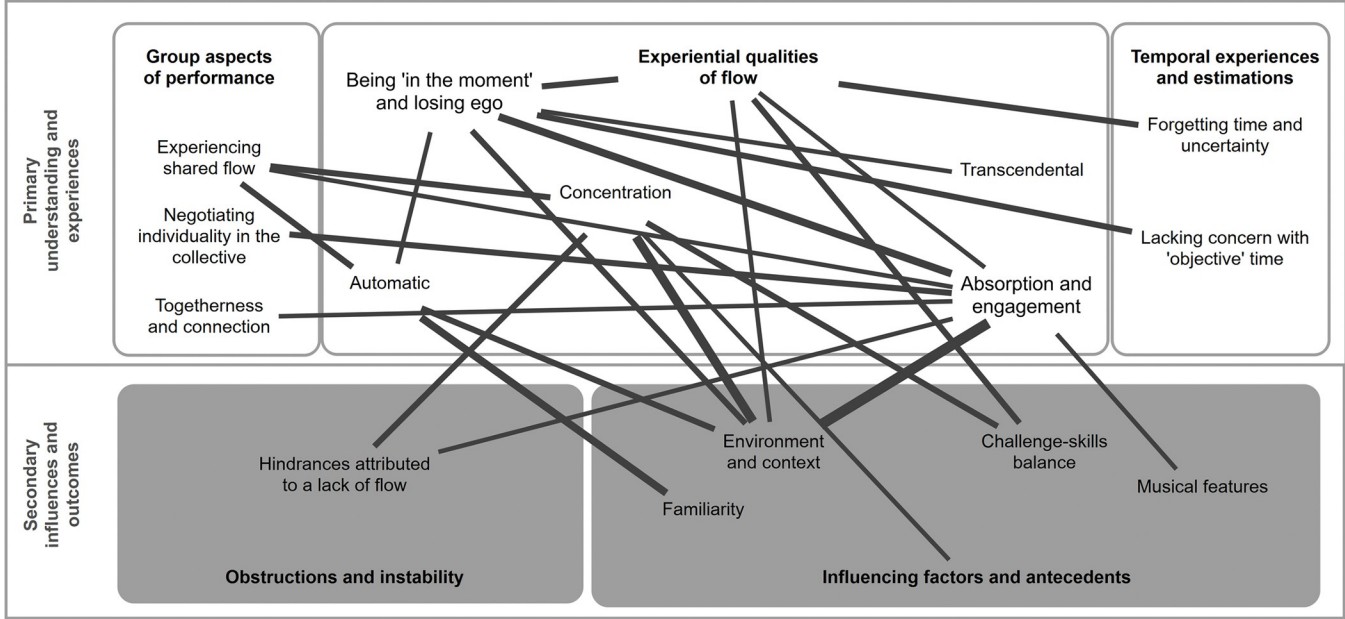

**Fig 3. Code interrelations for 'experiential qualities of flow' category and codes.** Only those codes that co-occur a minimum of three times are included in this figure. Line thickness denotes the number of co-occurrences present.

level of suggested consciousness that might differentiate them. A similar view was also associated with *being in the moment and losing ego* (1.3):

> "For me. It's like not having to think about what I'm playing or worry if I'm playing it right [. . .] like that point where you lose ego because actually worrying about what you're playing is an ego thing". (GSP, P4)

This sense of "being in the music" rested largely on the level of *familiarity* (5.2) a player had with the musical material. In addition to the level of *challenge-skill balance* (5.1) that must be in place for one to experience the absorbing aspects of flow, several noted on the level of *familiarity* (5.2), tied to flow occurring on an *automatic* (1.4) level. Examples include the following two quotes: "I think when you know the next thing that's going to happen, the next thing just follows". (SBGP, P2); and "I don't have to think about it. The structure, you just feel it. (NCH, P2)". These concepts were also tied to the kind of piece being performed and to the *musical features* (5.3) themselves.

The typical *musical features* (5.3) of gamelan, such as cyclicality and repetition were frequently attributed to the potential for *absorption and engagement* (1.1), and fundamental to the potential for flow to arise. This is explained as follows by one participant:

> "Cycling or there are some motions which have that repetitive or expected quality that you are expected to be able to, enter, a kind of I think when you know the next thing that's going to happen, the next thing just follows". (SBGP, P2)

Simultaneously, the same aspects of repetition may be regarded as *hindrances attributed to lack of flow* (4.1), often resulting in boredom depending on the level of *concentration* (1.5) and *challenge-skill balance* (1.1) in the moment. This is often specific to *instrumental roles* (5.4), as mentioned by one participant: "Mugirahayu is so easy for me I sometimes space out. So yeah, less challenge" (GSP, P2). Some players also found greater enjoyment in the improvisation as opposed to the traditional pieces. As the gong player put it: "it feels free for me to do what I want to do, [instead of] just waiting like after minutes after minutes and then falling asleep" (GSP, P3).

Nevertheless, participants often spoke of the experience of flow being a fine line with *absorption and engagement* (1.1), and *concentration* (1.5), whereby too much or too little of each would be seen as *hindrances attributed to lack of flow* (4.1). This was commonly dependent on the *environment and context* (5.5). On the topic of the improvisation, one participant stated "it's slightly more mentally engaged than what I think of as being flow. In that you are trying to find somebody, you're trying to find a sound to interact with" (SBGP, P1). As opposed to traditional pieces, where the goals are set from the start, this respondent felt that the level of conscious consideration and *concentration* (1.5) led to a loss of the potential for their playing to be *automatic* (1.4). In a similar vein, the level of challenge required for some parts during the traditional pieces led to overwhelm. This is captured well by the following statement: "I probably found the middle piece the hardest. So I don't think I was in flow state for that because I was a bit stressed because it's hard". (GSG, P1).

Interestingly, several participants when asked about their understanding of flow state, their explanation immediately surrounded a group context and *experiencing shared flow* (2.1). As one player admits "you're joining in with other people in it and you're all in this thing together" (SBGP, P1). Due to the intrinsic importance of the group in gamelan playing, it appears that it is difficult for participants to understand flow in an individual capacity in such a context: "And it's like it's a group thing that no individual is really in charge of." (NCH, P4).

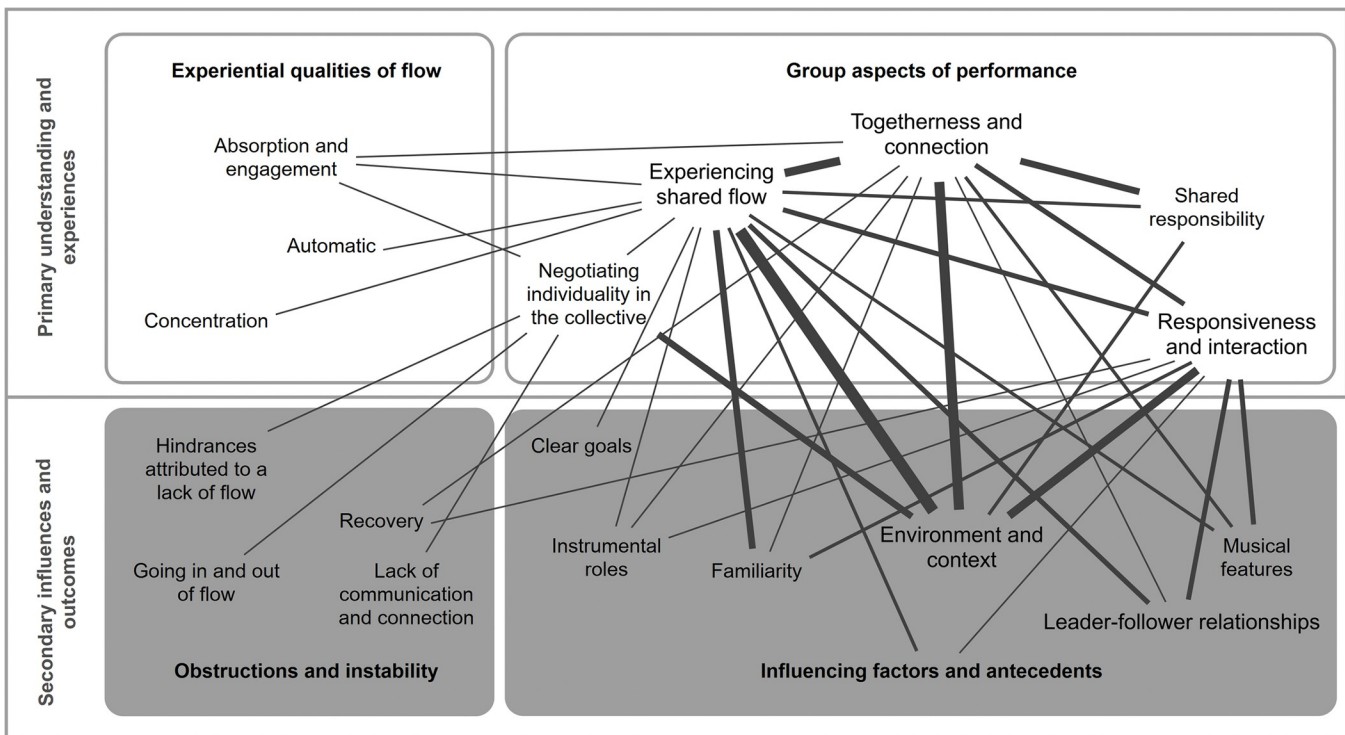

**Fig 4. Code interrelations for 'group aspects of performance' category and codes.** Only those codes that co-occur a minimum of three times are included in this figure. Line thickness denotes the number of co-occurrences present.

While several were aware of facilitating contexts and conditions that may lead to individual flow, one participant explained the complexities of group performance:

> "I would find it hard to describe individual flow in the context of what we just did there. I'd find it hard to say that I experienced flow outside of the group [. . .] it's very hard for musicians to say, I experienced flow, and it was only because of what I was doing. It was only within me, and it didn't involve any contamination from anybody else." (GSG, P2)

Accordingly, most groups felt that individual flow was far less applicable to their experiences of gamelan, and the focus turned to the group aspects of performance.

*2. Group Aspects of performance.* The experience of shared flow was discussed in much of the same terms as individual flow—including feeling absorbed, concentration, and performing automatically–though these could only occur when other aspects of the group were present. Fig 4 presents code interrelations between codes of this category and all others.

When the differences between group flow and individual flow were explained to participants, most groups differentiated between their *experience of shared flow* (2.1) in gamelan based on the degree of *togetherness and connection* (2.2) present, the level of *responsiveness and interaction* (2.3) they felt towards others of the group, and the need for *shared responsibility* (2.4) in the performance of both traditional and improvised playing. They recognised that experiencing individual flow in the context of gamelan as a *hindrance attributed to lack of flow* (4.1) overall, in that it disconnected them from the group in traditional pieces, leading them towards *negotiating individuality in the collective* (2.5).

One player attributed a sense of individual flow to getting lost, where their experience of *negotiating individuality in the collective* (2.5) and *going in and out of flow* (4.3) was a result of

excessive *absorption and engagement* (1.1) in their own part, yet eventually their recovery was prompted by a feeling of group *togetherness and connection* (2.2). This is stated explicitly by one participant:

> "I was really having fun with it on an individual level, because I knew exactly where I was and where everyone else was [. . .] then I was like [. . .] I don't know where I am now. That was definitely an individual flow state. I'm flowing with it, but sort of not necessarily in the right place [. . .] then when again that ciblon thing came back, there was this for me a kind of palpable sense of oh, group togetherness now". (SBGP, P1)

For this player in a group of collectively greater expertise, this occurred because they were at a level of gamelan musicianship where they were able to experiment with the material, regardless of it not being rehearsed. As with the general understanding of antecedents for flow, the importance of *familiarity* (5.2) was noted by several. As one player commented, with tradi- tional pieces, a group may not need to have much familiarity with one another providing there is a mutual awareness of the intention and *clear goals* (5.6) in mind:

> "It feels like it might be easier with, say, a traditional piece [. . .] where you could sit down with people that you've not played gamelan with before and probably fairly easily get into the same sense of flow because you know what it is you're aiming for." (GSP, P5)

Meanwhile, in certain *environments and contexts* (5.5) where the goal is less clear, such as in the improvisation, *familiarity* (5.2) among the group may hold more significance. One player noted that group flow may be more difficult to enter in an improvisation, "[unless] you've been playing together for 20 years and you kind of get into some groove in your improvisation that you're used to" (SBGP, P5). Across most groups, the improvisation seemed to be one in which players felt an increased sense of *negotiating individuality in the collective* (2.5), whereby they were more likely to have experienced individual flow, rather than group flow. This was attributed to the same sort of experimentation as that of the SBGP player, but also a *lack of communication* (4.4) and connection with others.

> "I felt in the improvisation, sometimes I was just in individual flow, doing my own little thing in a corner, not really caring what other people were doing. And other times I felt like I was playing with the other people and whatever was happening was happening. So I kind of felt I was kind of flipping between the two in the improvisation." (NCH, P4)

*Leader-follower relationships* (5.7) and *instrumental roles* (5.4) were closely tied *with togeth- erness and connection* (2.2), *responsiveness and interaction* (2.3), and the potential for *experiencing shared flow* (2.1) overall. With the exception of SBGP, the directors of the ensem- ble all commented on conscious considerations inhibiting their potential for flow.

> "[. . .] it was just the improvisation for me that I was able to sort of get into that flow state. Because for the other pieces [. . .] for me, personally, there were other very con- scious considerations going on in my mind. I was thinking about form and structure [. . .] whereas with the improvisation we didn't have to worry about what was right." (GSG, P2)

This pressure seemed to be particularly evident where there were less experienced members in the group, or members who were less experienced with their individual parts. The difference

with SBGP is that they are comprised entirely of highly experienced players, and therefore the leadership 'role' is changeable, perhaps most closely resembling the expert gamelan ensembles one would see in Java. The onus here was distributed across the group, and while they did not encounter the same pressures of singular leadership, there were still certain roles seen as crucial for others to rely on, which in turn had its own pressures that may inhibit flow, for instance, "if you're playing balungan . . . I believe you have to be absolutely kind of rigid [. . .] so it's a different kind of feeling of having to be there" (SBGP, P5).

As mentioned in relation to the category of *experiential qualities of flow* (1), excessive *absorption and engagement* (1.1) can be dangerous, and it is a fine line to draw. This becomes particularly problematic if they are fulfilling structural *instrumental roles* (5.4) such as the gong, or kenong, which tends to provide cues for other members of the ensemble. Musically, instruments such as the gong could be thought of as less demanding, and therefore less flow-inducing. Nevertheless, two participants of NCH argued for the contrary:

> "Could you get into a flow state if all you had to do was hit the gong? I would imagine that if you're part of a group and you're fully absorbed and everything, the group is happening [. . .] I would say you could be in the flow state if it was just a one note piece." (NCH, P1)

Theoretically, although the level of dextrous challenge may not be present for some instruments that would allow one to achieve the classic conception of flow, it is clear that such *instrumental roles* (5.4) still hold great importance, and the challenge is in how they provide the overarching form and structural significance. As another participant described, "it's a competence of listening, really. It's not a competence of playing the gong because that's easy. It's just knowing when to play it. (NCH, P4)". Having a sense of a *clear goal* (5.6) is important for *togetherness and connection* (2.2), which comes naturally in prescribed pieces, and was often attributed to the *lack of communication* (4.4) in improvisation. Overall, this category seemed to insist on the importance of the group above all else. Little mention of temporal perception was given through discussions of shared flow, but as has already been seen, in the context of gamelan, the conceptual overlap between the two is greatly significant. At the start, several defined flow itself as contributing to a loss of time sense, and in turn, participants commented that shared flow is more applicable to gamelan due to the need for interaction; it follows that the same temporal experiences may be present in shared flow.

*3. Temporal experiences and estimations.* The experience of temporal contraction is usually regarded as being closely tied to flow experience. Nonetheless, the focus group findings highlighted that *time flying* (3.2) was more related to influencing *factors and antecedents* (5), and the *environment and context* (5.5), than it was to qualities of flow. While groups were aware that a sense of time supposedly flies by when in flow, they more typically associated *forgetting time and uncertainty* (3.1) with flow, with comments such as "I think you forget time" (NCH, P2) and "I lose all track of time" (SBGP, P3). Code interrelations are presented in Fig 5.

When asked to reflect more on their estimates given during the experiment, however, many did not see the importance, *lacking concern with objective time* (3.3). One such participant stated: "It seems sort of irrelevant. I mean [. . .] I'm not concentrated on my perception of time whilst I'm playing. All that I'm really concerned with is you know, where we are with the structure [. . .] how it's progressing." (GSP, P1). This participant, along with several others, found that the nature of the questionnaire–in which they were requested to provide an estimate to the nearest minute and second following each performance–was trivial. Others found that *musical features* (5.3) provided some sort of temporal framework, although distinct from "objective" time, it separated the potential temporal consequences of flow experiences in music from other settings.

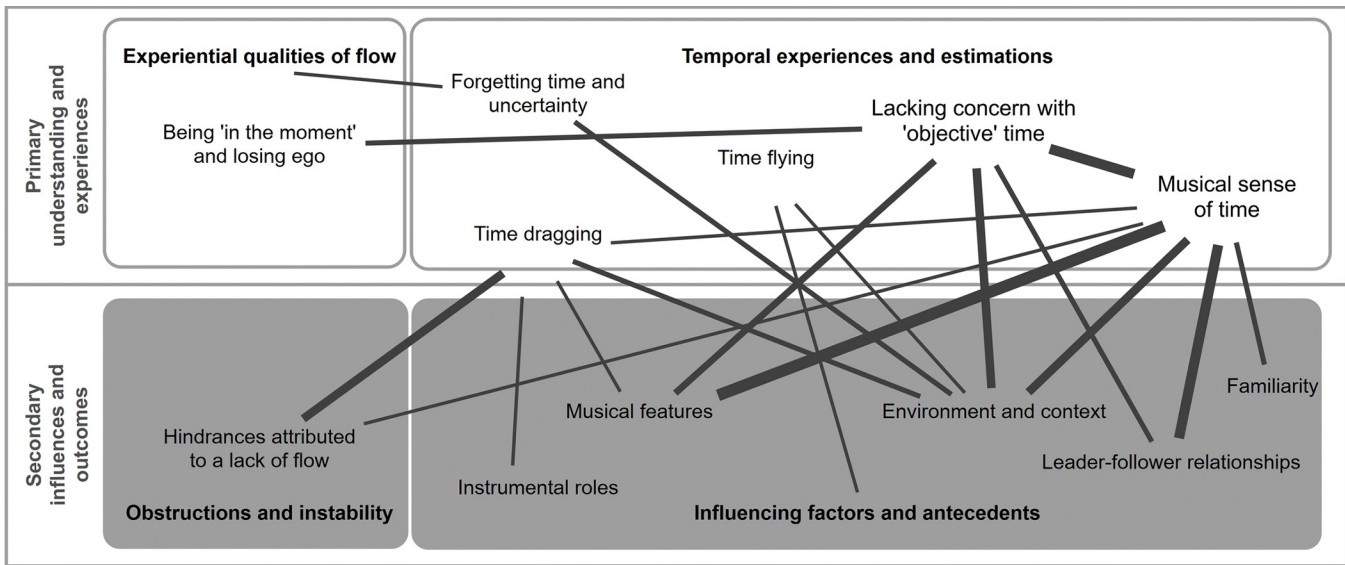

**Fig 5. Code interrelations for 'temporal experiences and estimations' category and codes.** Only those codes that co-occur a minimum of three times are included in this figure. Line thickness denotes the number of co-occurrences present.

"[with] flow state in other things, like, say, creative writing or doing art, it's a lot harder to keep track of time. I think with music, you've got a little bit of a framework in the back of your head. There's a beat and a pulse, and there's something there that helps you guesstimate how long it was." (NCH, P3)

For players who had roles that required leading tempo and structural changes, for instance, the drummer, the concept of metric, or a *musical sense of time* (3.4) was divorced from a sense of objective time even more:

"The thing is, when you talk about perception of time, if you're talking about how long the piece is, that's a conception of time that I think probably we don't have when we're playing, well I don't. But I am very concerned with time. So it's like the time between two beats. It's like, is there enough time for bonangs to be doing *imbal sekaran*? Is it like the optimum speed? So my brain is actually concerned with time constantly [...] metric time and how that relates to the instrumental parts of the feel I want." (GSP, P4)

This participant commented on the role of time in the sense of it unfolding, and the importance of their awareness of that. A similar point of view was provided when time was a consideration when it came to decision-making, related to *leader-follower relationships* (5.7):

"But there's another facet of that, which is as a piece of music progresses, you have to make decisions [...] so you can't entirely forget about time [...] Whenever you have to make a choice to change in a piece, even in a flow state, you have to be sensing that it's the right time because enough time has passed." (NCH, P1)

Together, the above comments demonstrate a difference in temporal awareness depending on whether they hold a leadership role. Even though they may been *lacking concern with objective time* (3.3), especially in flow, all group directors, or those who had experience in facilitating

groups and concerts had experience in timing pieces for concerts, and several presumed this knowledge would influence their estimates of how long a performance takes to play:

> "I think there's probably a difference between people who are sort of estimating how long the *lancaran* will take in this kind of situation or indeed, people who are used to timing how long a song is going to be [. . .] and you rely on that rather than actually being like, in this moment, I'm experiencing this much time." (SBGP, P2)

More experienced players, particularly those in SBGP, described that their estimates of the duration of pieces were made in relation to lengths of cycles in traditional pieces, which was also described as the reason the improvisations were much more difficult to estimate in terms of duration. Therefore, participants assumed that *familiarity* (5.2) and expertise seemed to help make more accurate assumptions, while such antecedents also lend themselves to flow. Accordingly, the resulting experience is opposing the presumed temporal effect of time distortion.

Several other influencing factors on time perception were much the same as those related to flow, though more in line with the expected effects. One player noted how strongly *hindrances attributed to lack of flow* (4.1), namely boredom, is tied to the idea of *time dragging* (3.5):

> "There's been moments when I've been playing gamelan and I'm not enjoying it. And I'm completely out of flow and there's a boredom factor comes in where time seems to drag for-ever [. . .] sometimes your flow does things to time, but so does boredom too. It's not the only thing that plays horrible games with your sense of time." (NCH, P4)

For some players and groups, the *environment and context* (5.5) were at the centre of this relationship. In particular, the improvisation was unanimously one in which there was a sense of *time dragging* (3.5), far more so than either of the traditional pieces, "When I first looked at it, it said estimate how long. My gut instinct was I wanted to write down three years" (GSP, P5). This may have been attributed to unfamiliarity with improvisation altogether, or lack of enjoyment and boredom.

Lastly, another influencing context, like those found in the previous two categories, was that of *instrumental roles* (5.4). This code commonly intersected with *time dragging* (3.5) but seemed to be a result of two opposing reasons. The first was related to complexity, where one player insinuated that time may have dragged on for those with more complex parts in a tradi-tional piece: "bonang panerus during the last of *inggah*. Really fast. He was like, 'When will this end?'" (GSP, P2). Conversely, for instruments that are less complex or have the greatest gaps in metric division, the same may also be true. This seemed apparent with gong players earlier on, while this was also commented on for the only balungan player in SBGP, where another player pointed out "you have the longest period between beats" (SBGP, P2), where they had been describing other reasons for time dragging.

Akin to that of flow experiences, the potential for *time flying* (3.2) is also a balancing act. One needs to be familiar and comfortable with their part, but not so familiar with timing that they can calculate an *irama* cycle to the nearest second. One needs a part that is challenging enough, that neither is extremely busy, nor extremely bare. But aside from all of this, the par-ticipants largely questioned the use in referring to objective measures of seconds and minutes at all.

**Secondary influences and outcomes.** Although many of the codes within the theme of secondary influences and outcomes have been discussed in their co-occurrences with other categories, we turn to consider the intersections between its two underlying categories:

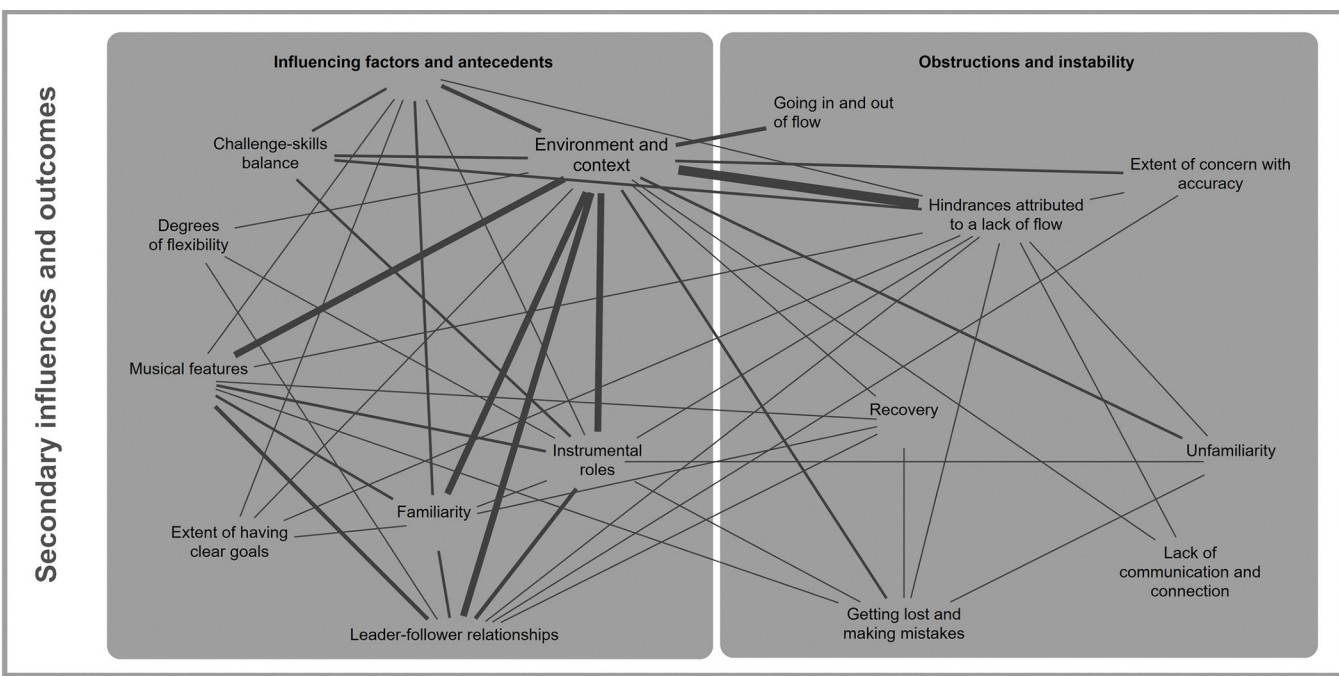

**Fig 6. Code interrelations between categories 'influencing factors and antecedents', and 'obstructions and instability' under the theme of 'secondary influences and outcomes'.** Only those codes that co-occur a minimum of three times are included in this figure. Line thickness denotes the number of co-occurrences present.

*influencing factors and antecedents (4)*, and *obstructions and instability (5)*. Code interrelations are shown in Fig 6.

Throughout all of the findings presented above, the *environment and context* (5.5) were key influences on temporal effects and the potential for flow experiences. Prescribed pieces, whether from notation or memory, were often related to *leader-follow relationships* (5.7) and *instrumental roles* (5.4), which in turn were ascribed to *hindrances attributed to lack of flow* (4.1). This was usually due to excessive pressure or responsibility for the group.

> "If I'm directing and the group isn't together, then I can't get into the flow at all. I've got to steer and give cues. [Kenong player] fell out at a bit. I gave a clear signal [. . .] he spotted it and got back in." (NCH, P1)

The improvisation, on the other hand, was one context in which participants could be free from such pressures, lending to *degrees of flexibility* (5.8). This was typically associated with how confined a participant felt to their part, and cooccurred with *instrumental roles* (5.4), *environment and context* (5.5), and *leader-follower relationships* (5.7): "with the improvisation we didn't have to worry about what was right. Yeah, so for me. That was the one. If I didn't have that [leadership] role, I think I'd have a different answer" (GSG, P2). *Degrees of flexibility* (5.8) for members with greater expertise, however, were also relevant to traditional pieces.

> "When we went into the change of drum, I was like, I don't know where I am now. That was definitely an individual flow state. I'm flowing with it, but sort of not necessarily in the right place, but in my own way." (SBGP, P1)

In such instances, *concern with accuracy* (4.5) was somewhat diminished, even if they were *getting lost and making mistakes* (4.6). Where improvisation lends itself to mutual, momentary decision-making, traditional pieces that have been rehearsed have a clear structure that can be relied upon. For GSP, the *unfamiliarity (4.7)* and lack of *clear goals* (5.6) in improvisation led towards contexts that were difficult to compare in terms of group flow, and potential *hindrances attributed to lack of flow* (5.1). One participant explained, "we haven't improvised together like that before, whereas we've practised Mugirahayu and Tukung a lot. So it's really tricky to compare things like group flow because you don't know what you're aiming for" (GSP, P4). Other participants described the other side of having such *clear goals* (5.6) in traditional pieces, especially where *leader-follower relationships* (5.7) were strongly tied to *instrumental roles* (5.4):

"[. . .] some of the instruments you feel you might not be as used to or are a bit more complicated when you start to learn it. And that can be harder to get into the group flow. Yeah, or else if you're doing the gong or something, you can't get lost. Like you really have to pay attention then." (NCH, P5)

The result of this was an excessive *concern with accuracy* (4.5), and the potential implications of *getting lost and making mistakes* (4.6).

Overall, some of the main determinants of flow experiences and temporal distortion, or lack thereof, were the environment and context, instrumental roles and leader-follower relationships, familiarity, and challenge-skills balance. Several of these factors can be accounted for by variables obtained through questionnaires. Accordingly, Study 2 aims to identify if similar trends and patterns can be identified through a combination of quantitative and qualitative data, using peri- and post-experiment questionnaires and surveys.

## Study 2

### Method

**Materials and measures.** Musical training was assessed within the pre-experiment questionnaire via the musical training subscale of the GOLD-MSI [35]. This questionnaire also encompassed demographics, questions related to their experience of gamelan playing in general, and four items on group flow antecedents [38], adapted for the context of group music making. These items were rated on a seven-point semantic differential scale and related to level of engagement, familiarity with the group, and perceived competence:

*(Pr1) I think the pieces today are (boring/engaging)*

*(Pr2) I know the other members of my group (not well/very well)*

*(Pr3) I think my own competence is (low/high)*

*(Pr4) I think the competence of my group was (low/high)*

After each piece, participants were asked to estimate how much time had passed while playing that piece and rate their confidence in their estimate. Interactive flow was assessed via the Interactive Flow (IF) questionnaire (Raettig & Weger, 2018), with some items reworded to suit the musical context (IF2, IF4, IF9, IF14) due to the validation context originally being that of a group discussion. This questionnaire featured 14 items, rated on a seven-point Likert scale from "completely disagree" to "completely agree":

*(IF1) We interacted like a well-rehearsed team.*

*(IF2) We had a stimulating performance.*

*(IF3) We felt the level of challenge was optimal.*

*(IF4) Our playing was fluid and smooth.*

*(IF5) We felt that time was flying by.*

*(IF6) We had no difficulty concentrating*

*(IF7) We had our wits about us.*

*(IF8) We were completely absorbed in what we were doing.*

*(IF9) We had a mutual understanding of our musical intentions.*

*(IF10) We always knew what we had to do next.*

*(IF11) We felt like we had everything under control.*

*(IF12) We forgot everything around us.*

*(IF13) Communication in our group went smoothly.*

*(IF14) We inspired each other.*

The *Interactive Flow questionnaire* encompasses 10 items of the Rheinberg instrument (Rheinberg et al., 2003), intending to measure underlying components of flow under factors of absorption and smoothness and reworded to refer to the group level of flow, while four additional items assessed the quality of the group's interactions (IF1, IF2, IF13, and IF14). This post-questionnaire was repeated after each of the three pieces, with the final section also encompassing the post-session outcomes designed by Raettig & Weger (2018), again adapted for the context with one item removed for lack of suitability. The four items were rated on a seven-point semantic differential scale:

*(Po1) I am (dissatisfied/satisfied) with the performance of my group*

*(Po2) Overall, the pieces we played were (boring/engaging)*

*(Po4) I think my own competence was (low/high)*

*(Po5) I think the competence of my group was (low/high)*

We also implemented the Inclusion of Other in Self scale [39], however, we do not discuss this here as this was a focus of our other study.

**Follow-up surveys.** For each group, several days after the study was conducted, participants were sent a personalised electronic survey. The aim of this was to gather reflections on potential differences between participants' estimations of time duration while playing, and the actual time. For each piece, participants completed two seven-point Likert scales, assessing the level of surprise they felt about the potential variation in perceived time estimates, and how similar they felt their estimates were to the rest of the group. An open-ended item then presented participants with each of the time estimates they provided during the experiment alongside the observed durations, and asked participants to account for the reasons for the potential difference between perceived and actual time, if there was any.

## Data analysis

**Quantitative measures.** All quantitative measures and analyses were calculated and carried out using R Studio version 4.3.1.

*Factor analysis.* To calculate latent variables underlying the Interactive Flow questionnaire [38], we conducted a factor analysis [33]. An initial confirmatory factor analysis was conducted on 105 overall observations in 36 participants of the IF questionnaire with three non-responses removed. The proposed model structure of "absorption", "smoothness", and "interaction" was not admissible for the observed data. We then proceeded with exploratory factor analysis, using maximum likelihood estimation, oblimin rotation, and parallel analysis based on factor analysis. The two-factor solution this yielded was subjected to a further confirmatory factor analysis using the highest loading items and maximum likelihood restricted estimation, while a cluster of cases was specified in the survey design to account for repeated data. The resulting solution consisted of two factors of 11 items, and reasonable fit for the observed data and small sample size (scaled $X^2$ (43, N = 105) = 75.169, robust CFI = 0.922, robust TLI = 0.901, robust RMSEA = 0.103, SRMR = 0.059) and very good internal consistency (Cronbach's $\alpha$ = 0.833 and $\alpha$ = 0.900). These two factors were labelled "interaction" and "absorption", to reflect the items related to the more interactive and group-dependent aspects of the IF questionnaire, and those that were more absorbing and automatic. Table 2 shows confirmatory factor loadings.

*Time distortion measures.* We adopted Im and Varma's approach to capturing time distortion [15] whereby the difference between participants' subjective time estimates and the objective performance time, measured by the experimenter based on video and audio recordings, was normalised by the objective performance time. This was calculated for each piece and each participant. The resultant values indicated underestimation (i.e., time flying) from negative values, and overestimation (i.e., time dragging) for positive values. Repeated measures mixed ANOVAs were used to first assess potential differences in time distortion and flow factors resulting from the playing condition or gamelan group alone, followed by Kruskal-Wallis and Wilcoxon tests to identify pairwise differences. Following this, we conduct preliminary Kendall's Tau correlations between the measure of time distortion across all conditions simultaneously, and all variables that may have influenced time perception according to the findings of Study 1. Graphical representation was created using a generalised pairs plot using the *GGally* package in *R* [40]. Correlations that appeared to be most prominent were incorporated into the building of a linear mixed-effects model using the *lme4* package [41, 42].

**Table 2. Confirmatory factor loadings for IF factors.**

| Factor | | Item | Estimate | Std. Err | z-value | p |
|---|---|---|---|---|---|---|
| **Absorption** | IF_2 | We had a stimulating performance | .854 | .128 | 7.971 | < .001 |
| | IF_3 | We felt the level of challenge was optimal. | .654 | .152 | 5.545 | < .001 |
| | IF_5 | We felt that time was flying by. | .680 | .142 | 6.573 | < .001 |
| | IF_8 | We were completely absorbed in what we were doing. | .615 | .101 | 6.632 | < .001 |
| | IF_12 | We forgot everything around us. | .624 | .132 | 7.150 | < .001 |
| | IF_14 | We inspired each other. | .714 | .127 | 6.509 | < .001 |
| **Interaction** | IF_1 | We interacted like a well-rehearsed team. | .829 | .108 | 10.518 | < .001 |
| | IF_4 | Our playing was fluid and smooth. | .811 | .140 | 7.959 | < .001 |
| | IF_9 | We had a mutual understanding of our musical intentions. | .843 | .171 | 7.128 | < .001 |
| | IF_11 | We felt like we had everything under control. | .754 | .145 | 8.950 | < .001 |
| | IF_13 | Communication in our group went smoothly. | .796 | .136 | 7.649 | < .001 |

N = 36, 105 observations.

While initial tests indicated non-normal distribution of all continuous variables, linear mixed-effects models are robust to such violations [43]. Therefore, we proceeded with a parsimonious approach to select an appropriate random effects structure of a model containing all hypothesised fixed effects. However, due to the limited sample, in all instances, the only error-free model arose through the inclusion of no random slopes and only a random intercept of participant nested within group. Following this, we reduced the fixed effects according to AIC criterion [44]. All model comparisons of random effects and subsequent fixed effects are provided in S1 and S2 Tables, while we only present the final model in this paper using *sjPlot* [45]. Where necessary, variables that involved total scores (i.e. IF pre and post scales, and GOLD--MSI musical training subscale) were scaled to improve model accuracy and balance amongst predictors.

**Follow-up survey analysis.**   A similar approach to that of the focus groups was taken with the follow-up surveys, to primarily capture insights related to any contextual influences on time estimates that may not have been described in focus groups. Insights were gained through elicitation, providing participants with their estimates of time passed against the actual observed timings, as opposed to being blind to the potential difference between perceived and actual as they were in the focus group presented in Study 1. Codes were assigned to this broader theme of contextual influences deductively, while an inductive approach allowed for unexpected insights. These codes were then used to investigate potential differences in self-reported time estimates and shared flow state based on the presence of the codes. Due to non-normal distributions amongst the outcome measures, and unequal variances between groups, we opted for Wilcoxon-Mann-Whitney tests using the *coin* package in *R* [46].

In the following section, we present the results as follows: we start with the questionnaire data, assessing relationships between time perception, shared flow factors, and variables identified to be of importance in accordance with the findings of Study 1. We then present mixed methods analyses between time perception and shared flow factors, and coded categories derived from follow-up surveys. The purposes of this were two-fold. Primarily, we sought to substantiate relationships identified via separate streams of qualitative and quantitative data, through a consolidated mixed-methods approach. Secondary to this, was to observe how well participants' reflections of potential influences on their estimates of time and experience of flow—when these were evident to participants, and gathered by means of elicitation—aligned with the influences identified from focus groups when participants were not aware of the differences between their perceived and objective measures of time. While through focus group interviews, several may have said their estimates were merely guesswork, the mixed methods findings could shed light on the potential for an unconscious timekeeping mechanism, and the factors that might interfere with this.

## Findings

**Questionnaire results.**   We used repeated-measures mixed ANOVAs, with Greenhouse-Geisser sphericity corrections, assigning time distortion, IF absorption, and IF interaction each as the dependent variable, condition as the within-subjects factor, and group as a between subjects factor to explain the potential variance attributed to the type of performance (i.e. traditional piece from notation or memory, or an improvisation) between groups. Kruskal-Wallis tests were then used to identify significant differences. Distributions of variables between groups and conditions, with pairwise significance between conditions derived from Wilcoxon rank sum and signed rank tests, are provided in Fig 7.

No significant effect of condition alone on time distortion was found, however, there were significant effects for group ($F(3,30) = 9.430$, $p < .001$), and for groups within conditions ($F$

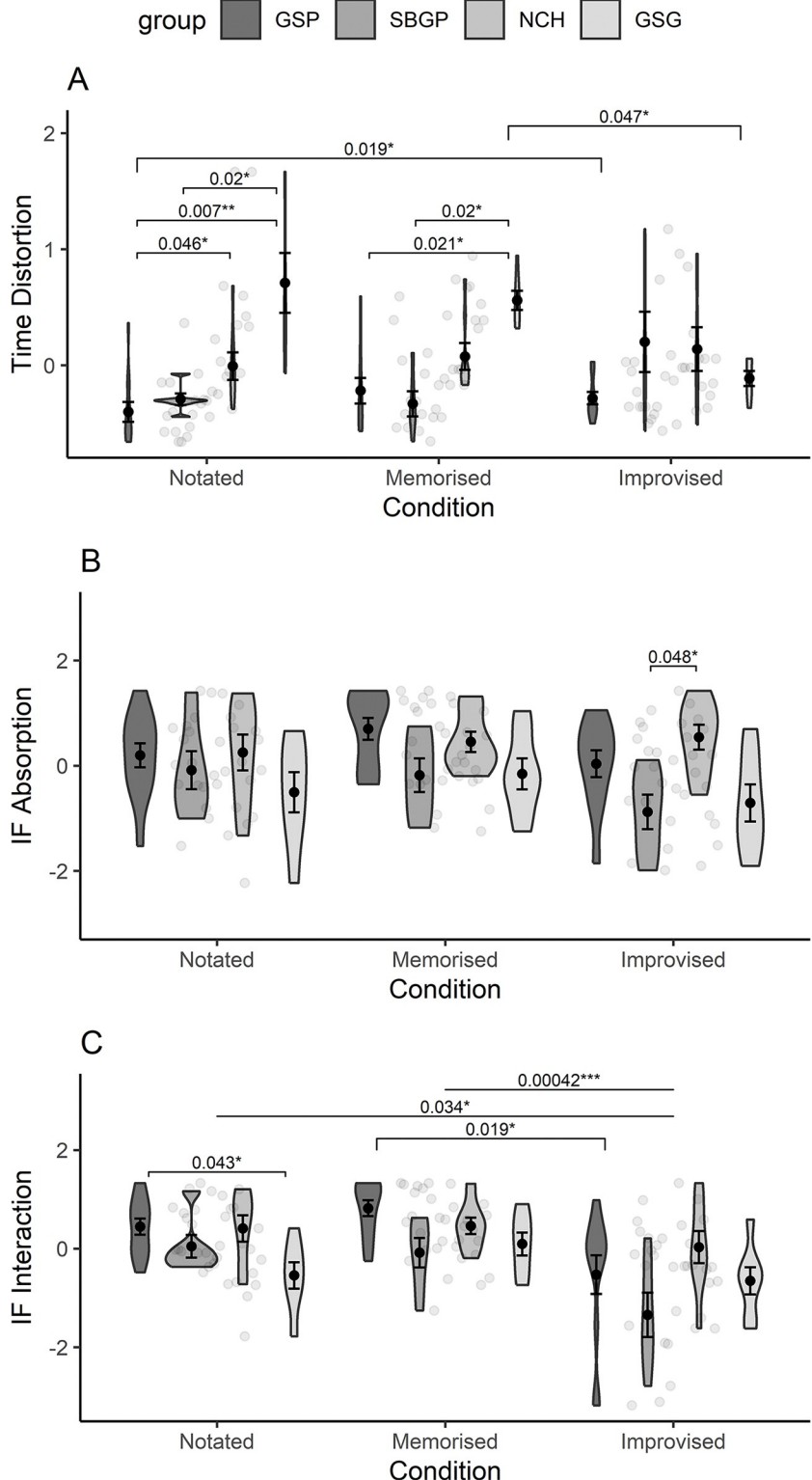

**Fig 7. Violin plots by group, split between playing conditions.** Distributions of A) time distortion, B) IF absorption, and C) IF Interaction. Error bars show the standard error of the mean. Note: *** Denotes p<001, ** p < .01, *p < .05,. p < .10.

(4.93,49.27) = 6.795, *p* < .001). Kruskal-Wallis tests revealed significant differences between groups for the notated (*H*(3,36) = 17.6, *p* < .001) and memorised conditions (*H*(3,35) = 16.3, *p* < .001), whereby generally GSP experienced significantly more time contraction in both conditions, while GSG experienced significantly more time expansion. GSG was the only group to demonstrate significant overall differences in time distortion between conditions (*H*(2,26) = 9.52, *p* = .009), whereby they reported time dragging the most in the memorised condition, compared to the improvised (see Fig 7).

For IF absorption, there were significant differences between groups (*F*(3,30) = 4.045, *p* = .016) but no main effect for conditions overall, nor for groups within condition. Follow-up Wilcoxon rank sum tests revealed GSG reported significantly lower absorption than both GSP (*W* = 546, *p* = .034) and NCH (*W* = 99, *p* = .015), while NCH reported significantly higher absorption than SBGP (*W* = 392, *p* < .007).

For IF interaction significant differences were found for both between groups (*F*(3,30) = 5.419, *p* = .004), and conditions overall (*F*(1.45,43.38) = 13.348, *p* < *.001*), There was no main effect between groups within condition. IF interaction was generally lower in the improvised condition than in the memorised (see Fig 7). As with IF absorption, GSG reported lower IF interaction compared to GSP (*W(21,36) = 572*, *p* = .008) and NCH (*W*(21,24) = 381, *p* < *.017*), while GSP reported higher IF interaction compared to SBGP (*W*(36,18) = 476, *p* = .033).

Preliminary correlations were examined between the measure of time distortion across all conditions simultaneously, and all variables that may have had an influence on time perception and flow, based on our review of the literature and focus group analyses. These involved, pre-, peri-, and post-experiment measures of interactive flow, GOLD-MSI musical training total score, years of gamelan playing experience. Of these, the only variables which demonstrated significant correlations with time distortion were the pre- and post-flow antecedents, and IF interaction. Fig 8 shows pairwise Kendall Tau correlation plots between all variables.

A linear mixed effects model was then fitted to predict time distortion, according to variables that correlated with time distortion or presented indications of a relationship without reaching the significance threshold, and those that participants described as being influencing factors on their perception of flow, and subsequently time in Study 1. Therefore, as well as assigning fixed effects of the IF Interaction factor and playing conditions to address the core of our aims, we incorporated musical training total scores and instrumental code according to metric density (i.e. elaborating instruments who played between the beats, melodic instruments who played every beat or more structurally punctuating instruments. This was contrast-coded to compare the lowest metric density against the middle and highest density, then to compare the middle and highest density against each other. We also incorporated IF absorption, as while the correlation was not significant, our results indicated differences between groups that may be controlled for through a random effect.

Table 3 presents the final model after a selection process reduced the fixed-effects structure based on AIC, while Fig 9 displays the fixed effects estimates graphically. Full details of all model comparisons are provided in S1 and S2 Tables.

IF interaction and IF absorption both significantly predicted time distortion in opposing ways. For the IF absorption scores, more time was perceived to have passed for those who scored more highly (estimate = 0.27, *p* = .004). On the contrary, for IF interaction, higher scores led to lower values of time distortion, interpreted as time contracting (estimate = -0.32, *p* = .001). Where condition was contrast coded, this effect was strongest in the notated condition, in comparison to both the memorised and improvised setting (estimate = 0.26, *p* = .012). However, planned contrasts with Benjamini-Hochberg adjustment for multiple comparisons did not reach significance when separately comparing the notated condition with the memorised (estimate = -0.28, *t*(87) = -2.05, *p* = .065) and improvised (estimate = -0.24, *t*(89.5) =

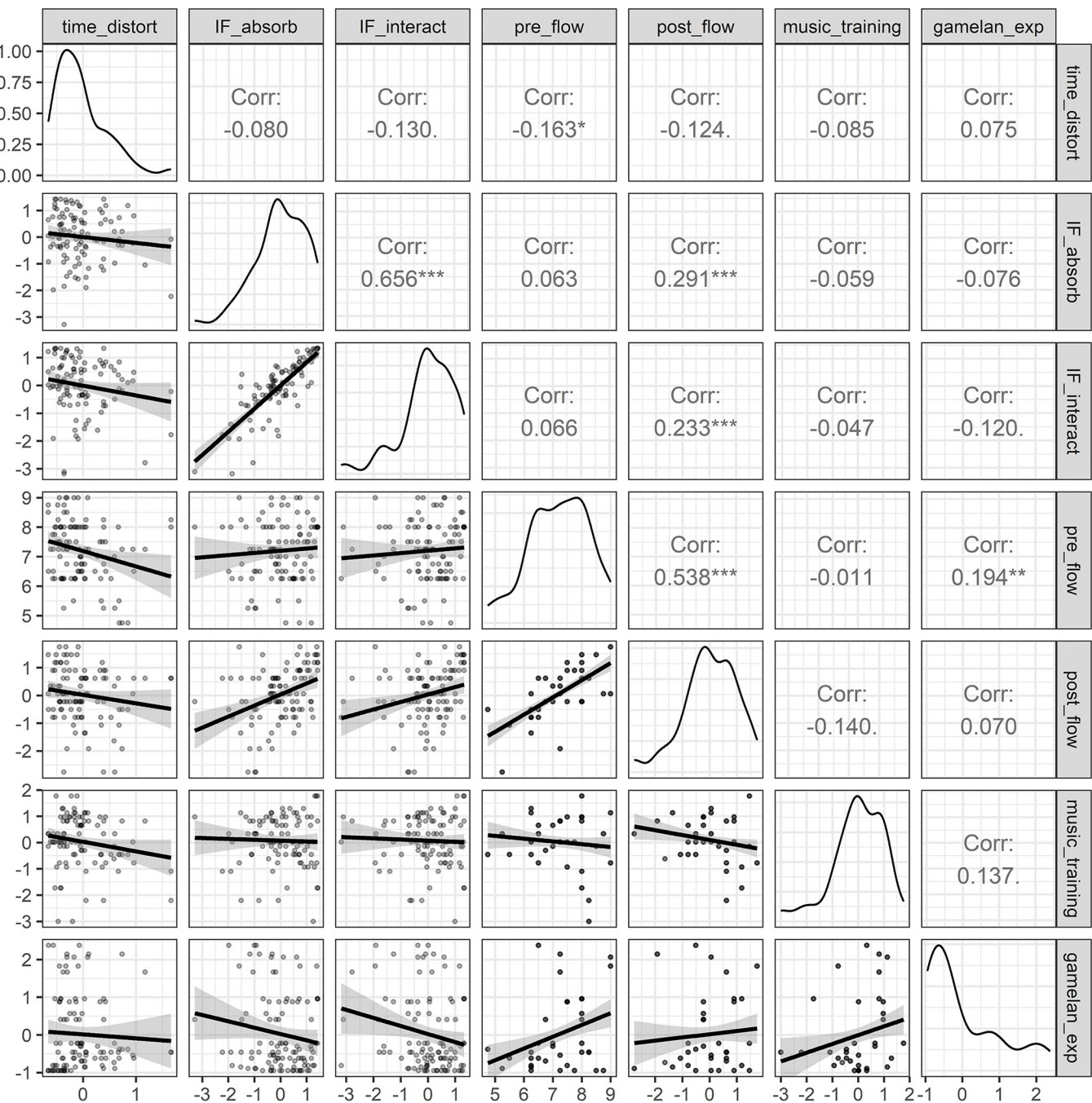

**Fig 8. Pairwise Kendall Tau correlations for time distortion, IF factors and antecedents, instrument grouping, and musical training.** The top right triangle shows Tau coefficients with significance. The diagonal represents variable distributions. The bottom left triangle shows scatter plots with smoothed regression lines and 95% confidence intervals. Note: *** denotes $p<001$, ** $p < .01$, *$p < .05$,.p < .10.

-2.118, $p = .065$) conditions. IF post-session outcome scores presented a comparable relationship with time contracting as with that of IF interaction (estimate = -0.15, $p = .042$).

Musical training also had a similar effect, where those with the highest level of musical training were more likely to have experienced a feeling of time flying (estimate = -0.22, $p = .029$. With regards to the contrast-coded instrumental roles, elaborating, metrically dense

**Table 3. Fixed and random effects estimates predicting time distortion, fitted with restricted maximum likelihood estimation.**

| Predictors | Time distortion | | |
| --- | --- | --- | --- |
| | Estimates | 95% CI | *p* |
| (Intercept) | -0.03 | -0.16 – 0.10 | 0.688 |
| IF interact | -0.32 | -0.52 – -0.12 | **0.002** |
| condition (notation vs without) | -0.06 | -0.23 – 0.11 | 0.507 |
| condition (memorised vs improvised) | -0.11 | -0.33 – 0.10 | 0.298 |
| IF absorb | 0.27 | 0.08 – 0.46 | **0.007** |
| Instrument (structural vs elaborating) | -0.25 | -0.48 – -0.01 | **0.040** |
| Instrument (balungan vs solo) | 0.08 | -0.15 – 0.31 | 0.510 |
| post flow | -0.13 | -0.27 – 0.00 | 0.057 |
| music training | -0.22 | -0.37 – -0.06 | **0.007** |
| IF interact X condition (notation vs without) | 0.26 | 0.05 – 0.47 | **0.018** |
| IF interact X condition (memorised vs improvised) | -0.05 | -0.30 – 0.21 | 0.714 |
| Random Effects | | | |
| $\sigma^2$ | 0.12 | | |
| $\tau_{00\ id\_n:group}$ | 0.08 | | |
| ICC | 0.40 | | |
| $N_{id\_n}$ | 33 | | |
| $N_{group}$ | 4 | | |
| Observations | 99 | | |
| Marginal $R^2$ / Conditional $R^2$ | 0.230 / 0.537 | | |
| AIC | 155.455 | | |

*Note*: A random effect was included for each group, and participant nested within group. Model formula: time distortion ~ IF interaction X condition + IF absorption + instrument + post flow + music training + (1|id:group).

instrumental roles led to greater time contraction compared to structural roles such as the gong (estimate = -0.25, *p* = .029). When carrying out planned contrasts to compare the effects of each instrument grouping, no statistically significant effect was found. It is worth highlighting that there was no significant effect of condition alone on the degree of time distortion felt.

**Mixed-methods findings.** For the final stage of analysis, we investigated whether participants' reflections, which arose through the follow-up surveys, could align with their estimates of time and self-reported shared flow experiences.

Through coding short segments of text provided in follow-up surveys, participants' assumptions as to the reasons behind their estimates were grouped according to the context. Wilcoxon-Mann-Whitney tests were used to examine potential significant differences in participants' self-reported variables based on dummy coded qualitative codes, i.e. whether the qualitative codes were present or not. Table 4 shows the most frequent reasoning codes, alongside the average quantitative variables for time distortion and IF factors for participants where such codes were present, and associated test results in the form of a joint-display table. Despite the limited sample, several significant findings emerged.

Within the notated condition, those who remarked that their parts require **excessive attention** levels seemed to significantly overestimate the time passed and were significantly more likely, alongside those who commented on **repetition**, to have had lower IF Interaction scores. Lower IF Absorption scores may have also related to both codes, however, they did not reach

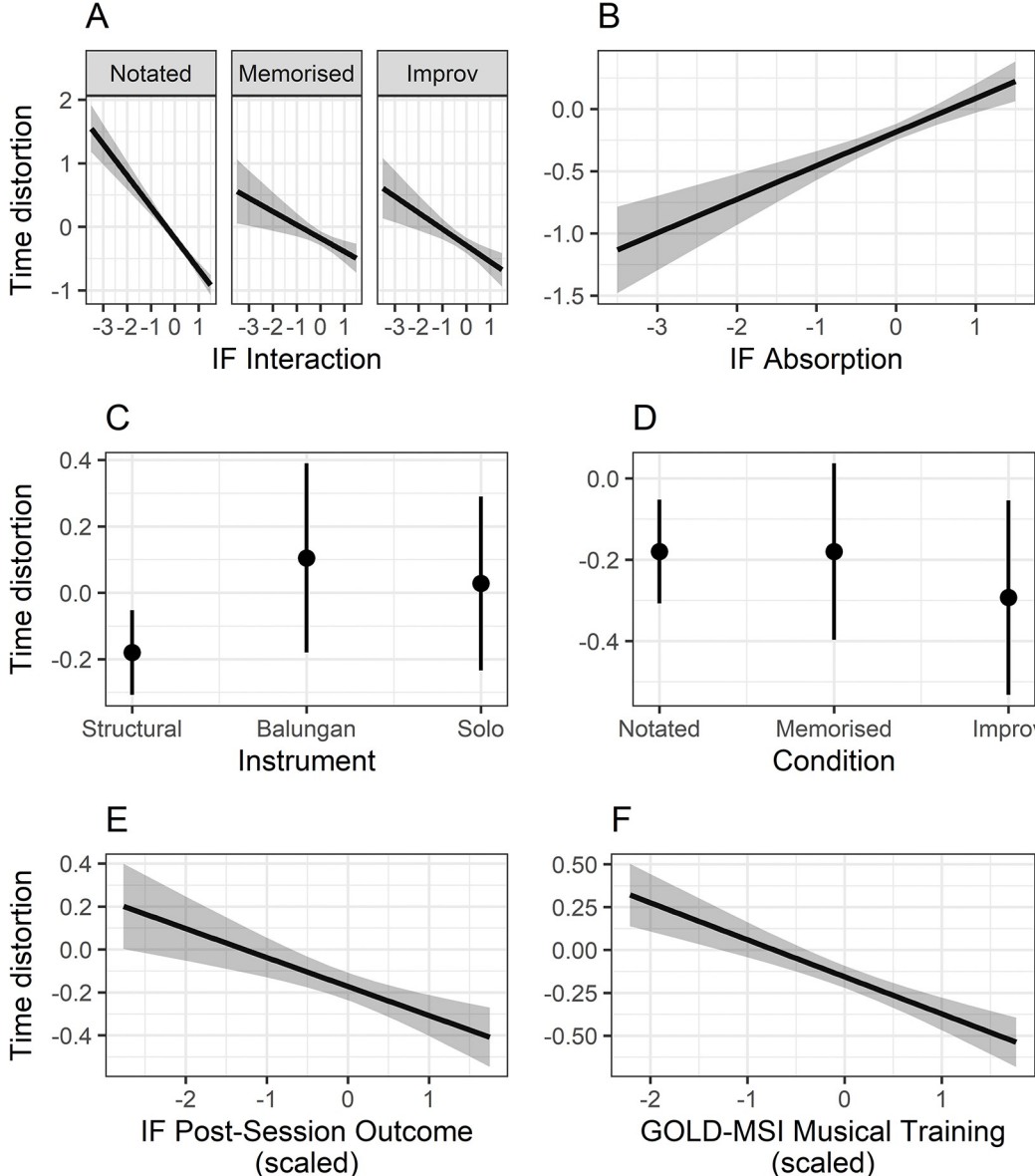

**Fig 9. Plots of each fixed effect estimates included in final linear mixed effects model presented in Table 1.** Fixed effects displayed are: A) Interaction effect of IF Interaction and Condition, B) IF Absorption, C) Instrument, D) Condition, E) IF Post-Session Outcome scores (scaled) and F) GOLD-MSI Musical Training subscale total scores (scaled). Shaded ribbons denote the standard error of the mean, while error bars show 95% confidence intervals.

the significance threshold. Participants who noted they had **prior awareness** of the amount of time a piece may have taken to play tended to significantly overestimate the amount of time. For comments related to **enjoyment**, the median of IF Absorption was greater than for those who did not report any enjoyable aspects, though this did not meet the significance threshold.

Regarding the memorised condition, the only finding that reached the significance threshold was for the **optimal challenge** code, whereby participants who commented on ideas relating to having a manageable task for their ability level underestimated the amount of time that had passed. While not meeting the significance threshold, the median IF absorption score for these participants was also greater than for those where this code was not present. Similarly to

the notated condition, participants who reported feeling they had held **excessive attention** levels may have overestimated the time passed on average, though this did not meet the significance threshold.

Few significant findings emerged in the improvised condition, participants whose comments related to **enjoyment** reported significantly higher IF absorption and IF interaction during the experiment. Players who made attempts at **listening and responding** to others throughout the improvisation reported higher absorption on average, both compared to those for whom the codes were not present, although this did not meet significance.

**Table 4. Joint display table for coded segments derived from follow-up surveys, compared with measures of time distortion, IF absorption and IF interaction, using Wilcoxon-Mann-Whitney tests.**

| Condition | Code | Exemplar quote | N | Present | Time distortion | | | | IF Absorption | | | | IF Interaction | | | |
|---|---|---|---|---|---|---|---|---|---|---|---|---|---|---|---|---|
| | | | | | Mdn | W | z | p | Mdn | W | z | p | Mdn | W | z | p |
| Notated | Optimal challenge | "I didn't feel like I was struggling [. . .] I was enjoying the music we were making." | 10 | 1 | -0.34 | 304 | 0.87 | .383 | 0.59 | 263 | -1.01 | .313 | 0.36 | 265 | -0.92 | .359 |
| | | | 19 | 0 | -0.20 | | | | -0.08 | | | | -0.02 | | | |
| | Enjoyment | "I was fairly relaxed and enjoying playing my part" | 8 | 1 | -0.29 | 317.5 | 0.12 | .903 | 0.82 | 276 | -1.90 | .057 | 0.80 | 282 | -1.61 | .107 |
| | | | 21 | 0 | -0.25 | | | | -0.18 | | | | -0.05 | | | |
| | Excessive attention | "I think this piece is quite extensive and it takes a lot more focus since I played kendhang." | 7 | 1 | **0.35** | **277** | **-2.70** | **.007** | -0.79 | 366 | 1.83 | .067 | **-0.68** | **381** | **2.60** | **.009** |
| | | | 22 | 0 | -0.32 | | | | .048 | | | | .56 | | | |
| | Repetition | "The piece is long though with many sections and repeats" | 7 | 1 | -0.15 | 304 | -1.33 | .185 | -0.35 | 364 | 1.73 | .083 | **-.30** | **375** | **-2.29** | **.022** |
| | | | 22 | 0 | -0.32 | | | | 0.48 | | | | .66 | | | |
| | Familiarity | "I probably lost track of time because Gambirsawit is another piece that I know well" | 7 | 1 | -0.33 | 345 | 0.76 | .444 | 0.65 | 316 | -0.71 | .476 | 0.81 | 302 | -1.43 | .154 |
| | | | 22 | 0 | -0.24 | | | | -0.01 | | | | 0.04 | | | |
| | Instrument Role | "When I play the suling I do struggle with time because it is out of sync that the notes are played" | 7 | 1 | -0.10 | 297.5 | -1.66 | .098 | -.18 | 341 | 0.56 | .575 | .70 | 326 | -.20 | .838 |
| | | | 22 | 0 | -0.31 | | | | .27 | | | | .08 | | | |
| | Prior awareness | "Usually when we play it, it clocks in around the three-minute mark" | 6 | 1 | **0.15** | **306.5** | **-2.074** | **.038** | 0.24 | 347 | 0.11 | .914 | 0.32 | 349 | 0.22 | .829 |
| | | | 23 | 0 | -0.31 | | | | 0.08 | | | | 0.10 | | | |
| Memorised | Familiarity | "This is a piece I am familiar with, and I really enjoy playing it" | 16 | 1 | -0.17 | 204 | 1.39 | .163 | 0.41 | 165.5 | -0.39 | .693 | 0.53 | 175.5 | 0.07 | .944 |
| | | | 12 | 0 | -0.08 | | | | 0.06 | | | | 0.36 | | | |
| | Prior awareness of timing | "I had recently timed the piece for our concert, so knew roughly how long it was (cheating, really!)." | 12 | 1 | -0.12 | 223 | -0.42 | .676 | 0.33 | 232 | 0 | 1 | 0.56 | 247 | 0.70 | .486 |
| | | | 16 | 0 | -0.24 | | | | 0.13 | | | | 0.43 | | | |
| | Optimal challenge | "I didn't experience any difficulties and was aware of everyone's part" | 10 | 1 | **-0.42** | **321** | **2.88** | **.004** | 0.60 | 224.5 | -1.75 | .080 | 0.64 | 232.5 | -1.37 | .172 |
| | | | 18 | 0 | -0.03 | | | | -0.01 | | | | 0.36 | | | |
| | Repetition and structure | "My part was very simple indeed and repetitive, just goes round and round so I wasn't thinking so much about it." | 9 | 1 | -0.31 | 293.5 | 0.89 | .375 | 0.62 | 273.5 | -0.10 | .922 | 0.59 | 250.5 | -1.23 | .219 |
| | | | 19 | 0 | -0.12 | | | | 0.20 | | | | 0.46 | | | |
| | Excessive attention | "I had a task that involved a high level of concentration, memory and decision-making, so my mind was very active." | 6 | 1 | **0.22** | **284** | **-1.96** | **.049** | -0.14 | 331.5 | 0.70 | .484 | -0.08 | 345.5 | 1.48 | .138 |
| | | | 22 | 0 | -0.24 | | | | 0.41 | | | | 0.56 | | | |

*(Continued)*

**Table 4.** (Continued)

| Condition | Code | Exemplar quote | N | Present | Time distortion | | | | IF Absorption | | | | IF Interaction | | | |
|---|---|---|---|---|---|---|---|---|---|---|---|---|---|---|---|---|
| | | | | | Mdn | W | z | p | Mdn | W | z | p | Mdn | W | z | p |
| Improvised[a] | Lack of direction and structure | "I didn't know what to do and thought the whole time I was waiting for the next part of the process what to do and how to work with others" | 19 | 1 | -0.13 | 295 | -.024 | .813 | -0.13 | 297 | 1.06 | .290 | -0.36 | 299 | 1.16 | .248 |
| | | | 10 | 0 | -0.16 | | | | 0.20 | | | | -0.04 | | | |
| | Active Engagement | "I was deliberately more active in the piece (I played the gong, therefore in the previous ones my activities were limited)" | 9 | 1 | -0.18 | 303 | 0.83 | .408 | 0.06 | 256 | -0.24 | .811 | -0.18 | 263 | 0.10 | .924 |
| | | | 20 | 0 | -0.09 | | | | 0.04 | | | | -0.35 | | | |
| | Enjoyment | "I was quite enjoying the improv, so it felt like it was over very quickly" | 8 | 1 | -0.23 | 326 | 0.54 | .591 | **0.70** | **242** | **-2.44** | **.015** | **0.16** | **249** | **-2.09** | **.037** |
| | | | 21 | 0 | -0.13 | | | | -0.10 | | | | -0.36 | | | |
| | Unfamiliarity | "We have never played like this before in this group so we were very unaware of how we would all play together." | 8 | 1 | -0.06 | 283.5 | -1.54 | .124 | -0.38 | 307 | 0.86 | .387 | -0.81 | 310 | 1.02 | .309 |
| | | | 21 | 0 | -0.26 | | | | 0.15 | | | | -0.23 | | | |
| | Listening and Responding | "My attention was so fixed on observing what others do and what I could contribute to the music" | 7 | 1 | -0.26 | 339.5 | 0.49 | .628 | 0.74 | 270 | -1.83 | .067 | -0.33 | 294 | -0.56 | .578 |
| | | | 22 | 0 | -0.13 | | | | -0.08 | | | | -0.34 | | | |

[a]*Note*: For the improvised condition, one less observation was available for the IF absorption and IF interactive factors due to missing data during the factor analysis. While there were 29 overall (28 in the memorised condition due to one participant leaving early), 1 less absent code than stated is the case for IF results in the improvisation.

## Discussion and conclusion

In this paper, we analysed qualitative perspectives alongside quantitative data, aiming to understand how the temporal effects of shared flow are experienced in the context of gamelan performance and participants' awareness of the factors that influence such effects. Gamelan is under-researched in music sciences, and as an inherently group-based musical setting, it provides the optimal conditions in which shared flow can arise and be studied. While this research is novel in this context, it also presents, to our knowledge, the first empirical enquiry into the temporal effects of shared flow in general.

### Participants' understanding of shared flow, time, and influencing conditions

Through our qualitative findings in Study 1, the temporal effects of flow seem to be common knowledge among participants, although a distorted sense of time was often treated as an outcome resulting from flow, rather than an experiential quality of flow itself. Flow was understood to be a state of complete absorption, whereby one loses a sense of self, is optimally challenged, and as a result, one loses a sense of time. Following these preliminary definitions, participants felt that no explicit distinction could be made between shared flow and individual flow in the context of gamelan, as by definition, success depends on the group. Moments where participants commented on feeling as though they had entered an individual sense of flow, particularly in the improvisation, seemingly coincided with moments where they felt detached from the group.

The conditions for temporal distortion also appear to be similar to that of shared flow; a middle ground between boredom and challenge, but also between novelty and experience. As

anticipated, many participants felt that assigning a numerical value to an estimate of how much time they felt had passed was arbitrary and that they rarely paid any attention to how long a piece took in minutes and seconds. For some, they stated it was a complete guess. For others, their estimates were based on cyclical patterns within the structure and the tempo underlying it, or on several occasions, prior knowledge of how much time a particular piece takes to play through based on timing for concert programming. While some participants believed that those with prior knowledge of timing pieces and an understanding of temporal structures would be more accurate in their estimates, our analyses in Study 2 demonstrated that this was not the case. Perhaps what is instead supported is the presence of a pace-maker accumulator model or internal clock [47, 48] or neural basis for the perception and estimation [49, 50], each affected by levels of arousal and attention. Even where participants felt that little could be gained from their estimations, this previous work seems to support deeper subconscious mechanisms that may have informed them.

## The influence of metric density and musical expertise, and dichotomous temporal effects of shared flow for temporal distortion

Using a linear mixed effects model, we acknowledged three key findings. Firstly, that time appeared to contract more for those with greater levels of musical training. Possible explanations for this are merely speculative but could have to do with greater familiarity with the material and increased potential challenge-skill balance to have been achieved, which has been shown to also relate to time flying in flow-related studies [13]. Secondly, we found that time flying was felt more for those playing structural instruments compared to the more metrically dense *balungan* or solo lines. This result was not surprising, and in fact, supports previous work related to the effect of metrical hierarchy, reduced cognitive load, and associated temporal effects [21, 51].

Thirdly, and perhaps one of the most fascinating revelations from our quantitative analysis, was a distinction between the relationship between time distortion and the two shared flow factors we assessed in this paper. The first factor, IF Interaction, which encompassed the more responsive and active elements of shared flow, seemed to present a negative relationship with measures of time distortion, and resulted in perceived time contracting. In turn, this effect was significantly stronger in the notated condition compared to others, potentially reflective of the importance of having clear goals for achieving shared flow [52]. Conversely, IF Absorption, relating to the more affective and absorbing qualities of shared flow, and curiously, an item related to time "flying by" presented a positive relationship with time distortion, resulting in a feeling of more time passing than actual. One potential reason, though purely conjecture, could be that IF Absorption is related more to feelings of individual flow. In focus groups this was discussed as leading to a lack of communication and concern for the group, aligning with the possibility of a mismatch between individual and group intentions [53]. While all items were worded with the use of plural pronouns, the IF Absorption factor seemed to most strongly encompass items that would be felt individually. Another theory this finding may lend support to is that shared flow may be experienced differently in accordance with the level of togetherness, attention, and self-other overlap present [54]. Further research is certainly needed to explore this possibility, and while we intended to account for multiple facets encompassing shared flow using the Interactive Flow questionnaire we used in this paper [38], the proposed factor structure was not an admissible fit for the observed data. As a result, our ability to assess the multi-dimensionality of shared flow was limited. We suggest future studies adopt a range of scales and methods, as we have aimed to do here, to try and understand how best shared flow experiences can be reported and measured.

Nevertheless, this study is the first to our knowledge to evidence a difference in how time is experienced according to the type of shared flow that is experienced. Where others before assessed the relationship between flow and the perception of time based on a singular measure of flow [15–17], our research supports the view that the temporal effects of flow are multi-faceted and complex, and should be treated as such.

## Attributed reasons for temporal distortion, and associated shared flow effects

Contrary to the old adage, "time flies when you're having fun", and prior work supporting this [55, 56], we found no significant relationship between those who attributed their durational estimates to enjoyment in the notated, nor improvised conditions. However, enjoyment in the improvisation did seem to be associated with higher levels of both IF Absorption and IF Interaction, and it seems that those who did enjoy the improvisation were far less vocal about it in the focus groups. Rather, many participants discussed the improvisation as one they enjoyed the least and in turn least identified with the concept of shared flow. This typically was attributed to having unclear goals, especially in comparison to that of the traditional pieces. In follow-up surveys, reasons for time estimates coded as lack of structure and direction, in fact, had no significant relationship with time distortion, nor IF factors. Nevertheless, participants who perhaps felt overwhelmed by their parts in the traditional pieces may have found the improvisation a welcome respite.

Attention and optimal challenge are frequently cited as defining characteristics of flow [12, 57], whether individual or shared [54], and in turn temporal contraction [15]. Mixed methods findings we reported support this, where reasons coded as optimal challenge in the memorised condition were significantly negatively correlated with temporal contraction. At the other end of the spectrum, participants who commented on feeling excessively challenged, or felt they had to focus very intensely, seemed to be significantly more likely to have reported an overestimation of time. In focus groups, several commented on the repetitive and cyclical elements of gamelan to be something that would encourage shared flow, but we observed a negative relationship between those who attributed their time estimates to repetition, and IF Interaction. In these instances, players may have felt under-challenged, bored or lost focus. Not only is this finding in keeping with flow theory, but it also echoes the theories of Locke and Thomas Reid [2], and is in keeping with ideas of complexity in music listening [10].

Our findings also may lend support towards the effects of directing attention towards time in music listening [10]. Many in focus groups also remarked that they estimated time based on their previous knowledge of the length of pieces and general structural durations, and accordingly, assumptions were made that the estimates of said players would be more accurate. However, in the notated condition, this code seemed to be associated with an overestimation of time, and players noted feeling surprised at the misalignment between perceived and objective time. These players may have experienced such slowing of time due to their attentional resources being directed towards time, especially when they were providing estimates in the second or third condition and were anticipating the question [15].

It is also worth highlighting that a key distinction between the findings of the focus group discussions in Study 1 and the findings of the follow-up surveys in Study 2 may be a result of potential differences between reasoning gained through speculation and elicitation, as was a key distinction in the methods. Just as methods of retrospective estimation or reproduction of timings seemingly rely on differing cognitive processes [6], so too might participants' views of what way influence time distortion vary in accordance with elicitation techniques or anticipatory reasoning. While observing the how reasoning of participants may differ based on methodological distinctions was not the focus of our research, this possibility arose through the

course of our analysis. Future studies may certainly wish to assess differences in reasoning under separate paradigms.

## Limitations

Despite the efforts made to maintain the naturalistic setting, inevitably the experimental context had several implications that were highlighted by participants as detracting from their usual experience. One such aspect was a lack of vocals. As a separate endeavour of the project involved collecting physiological data, they had been asked to omit vocals from any pieces they selected, so as not to create confounds in the signals. For less experienced groups in the West, this is less of an oddity, as it is common for pieces to be rehearsed and learnt almost entirely without vocals. However, for more advanced players, particularly those who have spent time in Java such as those in SBGP, vocals become much more prevalent, and at the forefront of much gamelan music. Furthermore, to the Javanese, much of the essence of what peak performances and the experience of shared flow may be attributed to is that of *rasa* [24, 58]. For native Javanese gamelan players, it is conceivable that wider cultural and societal considerations, and ideas of *rasa*, may be made if asked about factors that contribute to experiences of shared flow and time distortion.

Several limitations arose regarding the measures we used. As acknowledged by Im and Varma [15], when durational judgements are provided in a repeated-measures study, participants may treat prior estimates as an anchor towards their subsequent estimates. We sought to circumvent this effect by allowing groups to decide on the order of pieces they chose, and requesting they not look through the questionnaire before starting so they were not aware they would be asked how much time they thought had passed for the first condition at least. Regardless, three out of four groups decided to end with the improvisation, which may have influenced their temporal judgement in comparing pieces they had prior structural awareness of, with something that was completely unstructured.

Secondly, the use of the GOLD-MSI musical training subscale [35] was commented on several times by our participants as not reflecting the type of musical expertise they had with gamelan playing. One such item that caused some confusion was regarding how many instruments they played; as gamelan encompasses many different instruments, some participants chose to define all gamelan as one instrument, while others considered each gamelan instrument separately. Further to this, one may not spend hours of individual practice time with a gamelan instrument as one would with Western instruments, as it is only in the context of the group that some parts can be fully realised. For these reasons, this method of gaining data on musical expertise may not be fully appropriate in contexts outside of the Western musical tradition. As much of the groundwork in music psychology surrounds Western music [59], this limitation highlights the potential drawbacks of incorporating measures that have been validated with such a WEIRD focus. Of course, the participant pool in this research was largely Western even if the musical focus was not, and the consequence of the use of such a measure both in non-Western musical contexts and with non-Western participants could be all the more stark [60].

## Conclusion

So, how do musicians experience time when in shared flow? While typically characterised by feelings of time flying, in this paper we have unearthed a far more complex picture of flow-related temporal effects. It is true that temporal expansion and contraction may both relate to flow, but seemingly this entirely depends on the type of flow one is experiencing. The more active, and *inter*active qualities do seem to lend themselves to time flying, while the more absorbing, perhaps even individualised aspects of flow do not. Of course, many factors may influence time in the

same way they influence flow; challenge, arousal, and attention to name a few. All the while, we have also shown that the influence of each is a middle ground, akin to the inverted U-shape that flow is famous for, or indeed, similar to early philosophical accounts and findings related to music-listening experiences. It is still unclear whether flow leads to temporal distortion, or the other way around; or if such causality bears any significance. Nonetheless, we have shown that a quantitative measure of perceived time may be useful to understand the effects of flow, though little can be concluded through this alone. Future research may benefit from incorporating such a measure alongside self-reports and other methods to understand how truly interactive states of shared flow may differ from states that may be more individually experienced.

## Supporting information

**S1 File. Focus group protocol.** Semi-structured interview protocol for the four focus groups. (DOCX)

**S1 Table. Model comparisons of random effects.** Models are reduced in random effects in a stepwise manner while retaining all fixed effects. (DOCX)

**S2 Table. Model comparisons of fixed effects.** The most optimal model of S1 Table is reduced in fixed effects by means of AIC criterion improvement. (DOCX)

## Acknowledgments

We thank all members of the York Music Psychology group, including Noah Henry, Katherine O'Neill, Caroline Owen, Annalisa Mazzorali, and Serena Paese, and Adrian Kempf at the University of Graz for their invaluable support and feedback throughout all stages of the data collection and writing. We especially thank all groups that took part in the experiment, and for their enthusiasm and general interest in the research; Gamelan Sekar Petak, Southbank Gamelan Players, Dublin's National Concert Hall Gamelan, and Gamelan Spréacha Geala.

## Author Contributions

**Conceptualization:** Hannah J. Gibbs, Andrea Schiavio.

**Formal analysis:** Hannah J. Gibbs.

**Investigation:** Hannah J. Gibbs.

**Methodology:** Hannah J. Gibbs.

**Project administration:** Hannah J. Gibbs.

**Supervision:** Andrea Schiavio.

**Visualization:** Hannah J. Gibbs.

**Writing – original draft:** Hannah J. Gibbs.

**Writing – review & editing:** Hannah J. Gibbs, Andrea Schiavio.

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
