## [Decision Letter · Decision Letter 0]

13 Sep 2024

PONE-D-24-13763Flowing between gongs: mixed-methods insights into shared flow and temporal distortion in music performancePLOS ONE

Dear Dr. Gibbs,

Thank you for submitting your manuscript to PLOS ONE. After careful consideration, we feel that it has merit but does not fully meet PLOS ONE’s publication criteria as it currently stands. Therefore, we invite you to submit a revised version of the manuscript that addresses the points raised during the review process.

Dear author,

Thank you very much for submitting the article to the journal. Please read and refer to the reviewer's comments. After that we will be happy to receive the resubmission of the article.

We look forward to receiving your revised manuscript.

Kind regards,

Gal Harpaz, Ph.D.

Academic Editor

PLOS ONE

“At the time of publication, the first author is under receipt of AHRC Open-Competition Doctoral funding through the White Rose College of Arts and Humanities, grant number AH/R012733/1.”

3. We noted in your submission details that a portion of your manuscript may have been presented or published elsewhere. [This paper is part of a larger project, and therefore certain aspects of the procedure and methods, including participants, are the same as another currently in preparation. Due to distinct research questions and methodologies, we have maintained the papers as separate endeavours. We have stated this explicitly in this article, and we are happy to provide a copy of the work as a pre-print once it is ready for submission.] Please clarify whether this [conference proceeding or publication] was peer-reviewed and formally published. If this work was previously peer-reviewed and published, in the cover letter please provide the reason that this work does not constitute dual publication and should be included in the current manuscript.

Reviewers' comments:

Reviewer's Responses to Questions

**Comments to the Author**

1. Is the manuscript technically sound, and do the data support the conclusions?

Reviewer #1: Yes

2. Has the statistical analysis been performed appropriately and rigorously? 

Reviewer #1: Yes

3. Have the authors made all data underlying the findings in their manuscript fully available?

Reviewer #1: Yes

4. Is the manuscript presented in an intelligible fashion and written in standard English?

Reviewer #1: Yes

5. Review Comments to the Author

Reviewer #1: This research is very interesting, especially in trying to find the flow of gamelan players. Firstly, for understanding the qualitative data, there should be an exploration of cultural aspects because gamelan is not music in the same sense as understood by the Greeks. Almost all musical elements present in Western culture are different from gamelan, which does not have the concepts of intervals, melody, chord progression, harmony, improvisation, etc. The flow/durée in gamelan playing is very close to the everyday life of the Javanese, who are conflict-averse, along with the addition of playing experienced.

Secondly, Javanese gamelan is the sound behavior of the Javanese, more as a representation of conversation and cultural relations in a non-verbal way. In a gamelan group, all ways of life of the Javanese people can be found, and they usually place great importance on “rasa” (feeling). The essence of gamelan playing, whether achieving flow or peak experience, is entirely based on and considered according to “rasa,” which is also mentioned in Indian music.

Thirdly, for the quantitative aspect with repeated measures, it should be clarified further because the self-report used is different from the measurement scale.

6. PLOS authors have the option to publish the peer review history of their article (what does this mean?). If published, this will include your full peer review and any attached files.

Reviewer #1: **Yes: **DJOHAN

---

## [Author Response · Author response to Decision Letter 0]

19 Sep 2024

Editor comment 1. Please ensure that your manuscript meets PLOS ONE's style requirements, including those for file naming. The PLOS ONE style templates can be found at

https://journals.plos.org/plosone/s/file?id=wjVg/PLOSOne_formatting_sample_main_body.pdfand

Author response 1: We can confirm we have ensured that the manuscript adheres to PLOS ONE’s style requirements. You will note that the author affiliations have been amended, table formatting has been adjusted, and some headings have been changed, and all table and figure titles and legends have been amended. However, we do wish to keep the methods and results sections separate for the two studies for the sake of clarity, and other PLOS ONE papers seem to have done similar (e.g. https://doi.org/10.1371/journal.pone.0135646, https://doi.or g/10.1371/journal.pone.0304326). We hope this is acceptable.

As an aside, we have uploaded an alternative version of fig 2, as it was felt the formatting of this could be improved. The content of the figure has not changed.

Editor comment 2. Thank you for stating the following financial disclosure:“At the time of publication, the first author is under receipt of AHRC Open-Competition Doctoral funding through the White Rose College of Arts and Humanities, grant number AH/R012733/1.”Please state what role the funders took in the study. If the funders had no role, please state: "The funders had no role in study design, data collection and analysis, decision to publish, or preparation of the manuscript." If this statement is not correct you must amend it as needed. Please include this amended Role of Funder statement in your cover letter; we will change the online submission form on your behalf.

Author response 2: This is correct, the funders had no role in the study design. We have included the amended statement in the cover letter

Editor comment 3. We noted in your submission details that a portion of your manuscript may have been presented or published elsewhere. [This paper is part of a larger project, and therefore certain aspects of the procedure and methods, including participants, are the same as another currently in preparation. Due to distinct research questions and methodologies, we have maintained the papers as separate endeavours. We have stated this explicitly in this article, and we are happy to provide a copy of the work as a pre-print once it is ready for submission.] Please clarify whether this [conference proceeding or publication] was peer-reviewed and formally published. If this work was previously peer-reviewed and published, in the cover letter please provide the reason that this work does not constitute dual publication and should be included in the current manuscript.

Author response 3: This is correct, and we are glad to clarify. We have amended the statement we previously provided in the cover letter as follows: “This paper is part of a larger project, and therefore certain aspects of the procedure and methods, including participants, are the same as another currently in preparation. Due to distinct research questions and methodologies, we have maintained the papers as separate endeavours. The other paper is currently in review with another journal, and the pre-print is now available here: https://doi.org/10.21203/rs.3.rs-4581247/v1. We have stated this explicitly in this article. The other paper details the relationship between shared flow and physiological measures, self-other overlap, and an alternative approach of time perception, being that of consensus among the group. To include all aspects of this large-scale study within the same publication would not be possible, and for that reason these two papers have been designed as distinct from one another. This does not constitute dual publication”.

Editor comment 4. When completing the data availability statement of the submission form, you indicated that you will make your data available on acceptance. We strongly recommend all authors decide on a data sharing plan before acceptance, as the process can be lengthy and hold up publication timelines. Please note that, though access restrictions are acceptable now, your entire data will need to be made freely accessible if your manuscript is accepted for publication. This policy applies to all data except where public deposition would breach compliance with the protocol approved by your research ethics board. If you are unable to adhere to our open data policy, please kindly revise your statement to explain your reasoning and we will seek the editor's input on an exemption. Please be assured that, once you have provided your new statement, the assessment of your exemption will not hold up the peer review process.

Author response 4: Originally the link to the OSF repository was set to read-only rather than public for the sake of the review, while we ensured that all materials were finalised. This has now been completed, and so the repository has been made public. The doi for the repository is here: DOI 10.17605/OSF.IO/TV7X9, and the link is here: https://osf.io/tv7x9/. This box indicating that the DOI would be available after acceptance should now be unticked, and the data availability statement should be amended with this link.

Editor comment 5. Your ethics statement should only appear in the Methods section of your manuscript. If your ethics statement is written in any section besides the Methods, please delete it from any other section.

Author response 5: The ethics statement has now been retracted from the bottom of the manuscript, and remains only in the methods section.

Editor comment 6. Please include captions for your Supporting Information files at the end of your manuscript, and update any in-text citations to match accordingly. Please see our Supporting Information guidelines for more information: http://journals.plos.org/plosone/s/supporting-information.

Author response 6: The captions for supporting information files and the in-text citations for them have been updated accordingly.

Editor comment 7. Please review your reference list to ensure that it is complete and correct. If you have cited papers that have been retracted, please include the rationale for doing so in the manuscript text, or remove these references and replace them with relevant current references. Any changes to the reference list should be mentioned in the rebuttal letter that accompanies your revised manuscript. If you need to cite a retracted article, indicate the article’s retracted status in the References list and also include a citation and full reference for the retraction notice.

Author response 7: We can confirm that the reference list is complete and correct. Where any errors were noted, these have been amended, and all have been formatted to adhere to APA style. Two references have been added to the list in order to address the comments of the reviewer:

Becker J. Deep listeners: Music, Emotion, and Trancing. Bloomington: Indiana University Press; 2004.

Sawyer K. Group Creativity: Music, Theater, Collaboration. Oxford: Taylor & Francis Group; 2003.

Reviewer comments:

Reviewer comment 1: This research is very interesting, especially in trying to find the flow of gamelan players. Firstly, for understanding the qualitative data, there should be an exploration of cultural aspects because gamelan is not music in the same sense as understood by the Greeks. Almost all musical elements present in Western culture are different from gamelan, which does not have the concepts of intervals, melody, chord progression, harmony, improvisation, etc. The flow/durée in gamelan playing is very close to the everyday life of the Javanese, who are conflict-averse, along with the addition of playing experienced.

Author response 1: We agree that greater detail in cultural aspects of gamelan playing may be useful to the reader, particularly in describing the relevance of improvisation, which is not traditional in gamelan per se. For that reason we have provided said detail in the introduction [line 123-125] and limitations sections [line 938-940] of the paper. We wish to note that in this case, the participants were Western musicians, and the gamelan ensembles are based in the UK and Ireland. While most British and Irish gamelan ensembles aim to instil many of the attitudes of Javanese gamelan, many Western players will still approach the learning and performance of gamelan with a Western musical lens. 

Reviewer comment 2: Secondly, Javanese gamelan is the sound behavior of the Javanese, more as a representation of conversation and cultural relations in a non-verbal way. In a gamelan group, all ways of life of the Javanese people can be found, and they usually place great importance on “rasa” (feeling). The essence of gamelan playing, whether achieving flow or peak experience, is entirely based on and considered according to “rasa,” which is also mentioned in Indian music.

Author response 2: We thank you for highlighting this very important point. Initially we felt that delving into ideas of “rasa” may be overly exhaustive in the case of this paper, but we appreciate this opportunity to explore this deeper. A short explanation of rasa and its relevance to flow has now been included in the background section [line 118-122] and limitations section [line 936-938].

Reviewer comment 3: Thirdly, for the quantitative aspect with repeated measures, it should be clarified further because the self-report used is different from the measurement scale.

Author response 3: Thank you for noting this. We had stated that some items of the IF were reworded to suit the context, but we have now provided greater specificity [lines 578-579]. Furthermore, we have added some additional clarity as to why the factors differ from the proposed structure [lines 597 and 626-627].

---

## [Decision Letter · Decision Letter 1]

8 Nov 2024

Flowing between gongs: mixed-methods insights into shared flow and temporal distortion in music performance

PONE-D-24-13763R1

Dear Dr. Gibbs,

We’re pleased to inform you that your manuscript has been judged scientifically suitable for publication and will be formally accepted for publication once it meets all outstanding technical requirements.

Kind regards,

Gal Harpaz, Ph.D.

Academic Editor

PLOS ONE

Additional Editor Comments (optional):

Thank you for submitting the revised article to the journal. After receiving the reviewer's comments on the revised version of the article, the article can be accepted for publication.

Reviewers' comments:

Reviewer's Responses to Questions

**Comments to the Author**

1. If the authors have adequately addressed your comments raised in a previous round of review and you feel that this manuscript is now acceptable for publication, you may indicate that here to bypass the “Comments to the Author” section, enter your conflict of interest statement in the “Confidential to Editor” section, and submit your "Accept" recommendation.

Reviewer #2: All comments have been addressed

2. Is the manuscript technically sound, and do the data support the conclusions?

Reviewer #2: Yes

3. Has the statistical analysis been performed appropriately and rigorously? 

Reviewer #2: I Don't Know

4. Have the authors made all data underlying the findings in their manuscript fully available?

Reviewer #2: Yes

5. Is the manuscript presented in an intelligible fashion and written in standard English?

Reviewer #2: Yes

6. Review Comments to the Author

Reviewer #2: The writers have adequately responded to the reviewer's comments and made the requested changes - according to the comments as presented to me.

7. PLOS authors have the option to publish the peer review history of their article (what does this mean?). If published, this will include your full peer review and any attached files.

Reviewer #2: **Yes: **Tal Vaizman

---

## [Editor Report · Acceptance letter]

12 Nov 2024

PONE-D-24-13763R1 

PLOS ONE

Dear Dr. Gibbs, 

I'm pleased to inform you that your manuscript has been deemed suitable for publication in PLOS ONE. Congratulations! Your manuscript is now being handed over to our production team.

Kind regards, 

on behalf of

Dr. Gal Harpaz 

Academic Editor

PLOS ONE